# THE BEST DEFENSE IS ATTACK: REPAIRING SEMANTICS IN TEXTUAL ADVERSARIAL EXAMPLES

## ABSTRACT

Recent studies have revealed the vulnerability of pre-trained language models to adversarial attacks. Existing adversarial defense techniques attempt to reconstruct adversarial examples within feature or text spaces. However, these methods struggle to effectively repair the semantics in adversarial examples, resulting in unsatisfactory performance and limiting their practical utility. To repair the semantics in adversarial examples, we introduce a novel approach named Reactive Perturbation Defocusing (`Rapid`). `Rapid` employs an adversarial detector to identify pseudo-labels for adversarial examples and leverage adversarial attackers to repair the semantics in adversarial examples by adversarial attacks. Our extensive experimental results, conducted on four public datasets, spanning various adversarial attack scenarios, convincingly demonstrate the effectiveness of Rapid. To address the problem of defense performance validation in previous works, we provide a demonstration of adversarial detection and repair based on our work, which can be easily evaluated at `https://tinyurl.com/22ercuf8`.

## 1 INTRODUCTION

Pre-trained language models (PLMs) have achieved state-of-the-art (SOTA) performance in a variety of natural language processing tasks (Wang et al., 2019a;b). However, recent studies (Li et al., 2019; Garg & Ramakrishnan, 2020; Li et al., 2020; Jin et al., 2020; Li et al., 2021; Boucher et al., 2022) showed that PLMs are highly susceptible to adversarial examples, **a.k.a. adversaries**, created by subtly changing the selected words in a natural examples (a.k.a., clean examples) (Morris et al., 2020). Despite a widespread acknowledgment of the critical importance of adversarial robustness in the deep learning community (Alzantot et al., 2018; Ren et al., 2019; Zang et al., 2020; Zhang et al., 2021; Jin et al., 2020; Li et al., 2021; Wang et al., 2022a), research dedicated to textual adversarial defense remains comparatively underexplored compared to the field of computer vision (Rony et al., 2019; Gowal et al., 2021; Wang et al., 2023; Xu et al., 2023). Existing works for textual adversarial defense can be mainly classified into adversarial training-based (Liu et al., 2020a;b; Ivgi & Berant, 2021; Dong et al., 2021b;a) and reconstruction-based (Zhou et al., 2019; Jones et al., 2020; Bao et al., 2021; Keller et al., 2021; Mozes et al., 2021; Li et al., 2022; Shen et al., 2023) approaches.

The crux of existing adversarial defense studies is that they cannot precisely distinguish the semantic differences in natural and adversarial examples, let alone repair the semantics in adversaries. In other words, the existing adversarial defense methods fail the guarantee semantic similarities between natural examples and repaired examples. We provide an example to illustrate this phenomenon in Figure 1. In this example, it is observed that `RS&V` (Wang et al., 2022b), one of the latest adversarial defense works, cannot model the semantic differences in adversarial and repaired examples. This is because augmentation-based defense methods are untargeted and do not truly learn to eliminate adversaries. Another significant problem is that many existing studies (Mozes et al., 2021; Wang et al., 2022b) are unable to efficiently pre-detect adversaries before the defense process. These approaches indiscriminately treat all input texts, disregarding the necessity of discerning between adversaries and natural examples for an optimal defense strategy. This obstacle not only incurs a waste of computational resources but also results in an unnecessary defensive posture on natural examples, potentially exacerbating performance degradation.

Keeping the above-discussed two challenges in mind, we propose a novel paradigm for adversary defense. Firstly, we train an effective adversarial detector along with the victim model to achieve

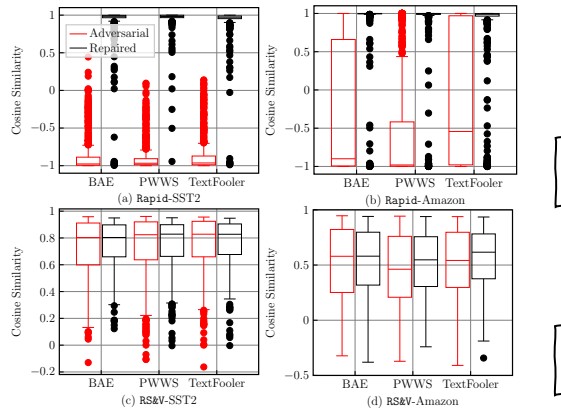

Figure 1: The visualizations of semantic feature-based cosine similarity on binary classification datasets. The "Adversarial" and "Repaired" denote the cosine similarity distribution between adversarial-natural example pairs and repaired-natural example pairs, respectively. The visualizations are from `Rapid` and `RS&V` (Wang et al., 2022b). respectively.

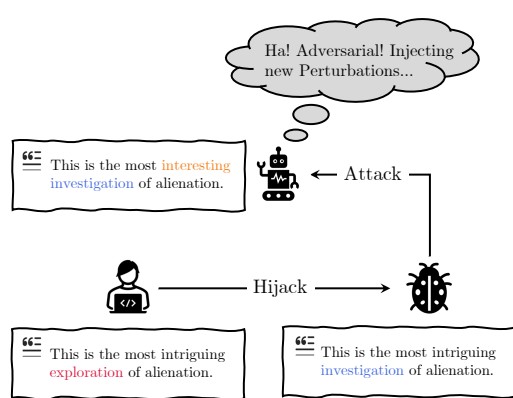

Figure 2: An example of reactive perturbation defocusing in sentiment analysis. `Rapid` will not apply defense unless an adversary is detected. The original word in this example is exploration. Perturbation defocusing repairs the adversary by introducing safe perturbations (interesting) to distract the objective model from the malicious perturbation (i.e., investigation).

fast 'in-victim-model' adversary detection, i.e., parallel adversary detection with standard modeling (e.g., classification) of the victim model. In this way, we can pre-detect adversaries and alleviate unnecessary repair processes on natural examples. Secondly, we propose a new adversary repair method called reactive `Rapid`, which is based on adversarial attacks. Overall, our adversary defense paradigm is characterized by the following three key aspects:

**Adversarial Semantics Repair**. Compared to augmentation-like methods, `Rapid` leverages adversarial attackers to counterattack the adversaries, injecting a few safe perturbations into adversaries to distract the victim model from malicious perturbations. Due to the principle of minimizing edits (in this work, we refer to the semantics in adversaries as the features encoded by PLM for simplicity), we can mitigate semantic shifts in repaired examples. The semantics of the repaired examples encoded by the victim model are very similar to the original, natural examples, as evidenced by Figure 1. Additionally, more examples can be viewed in the online click-to-run demo. We call this method "perturbation defocusing" (`PD`). It aims to defocus the malicious perturbations to repair the deep semantics instead of leading to further semantic shifts.

**Reactive Adversarial Defense**. `Rapid` uses the adversarial detector to concentrate its defensive efforts primarily on pre-detected adversaries, thus minimizing any collateral impact on natural examples (Xu et al., 2022) and reducing resource waste in defending against natural examples. As illustrated in Figure 2, `PD` introduces new perturbations into the adversary to create a repaired example that repairs the previous mis-prediction of the adversary.

**In-victim-model Adversarial Detector**. To overcome the limitations associated with inefficient adversary detection, `Rapid` builds an 'in-victim-model' adversarial detector, a binary classifier grounded in PLM architecture. The adversarial detector is jointly trained with the victim model as a multitask modeling task, allowing it to detect adversaries with no additional cost. In practice, this adversarial detector can recognize adversaries produced by a range of attackers, thanks to its training on a diversified set of adversaries from multiple adversarial attackers.

In conclusion, here are some findings from our experiments.

a) We illustrate a counterintuitive fact that adversarial attacks are efficient in repairing the adversaries. `Rapid` can maintain the deep semantics in repaired examples, which has been ignored in previous research and is vital for adversarial defense.

b) `Rapid` use a pseudo-similarity filtering strategy to select repaired examples that can achieve up to $99.9\%$ repair accuracy of adversarial examples on binary classification datasets, significantly surpassing text/feature-level reconstruction and voting-based methods.

c) We present a simple but effective in-victim-model adversarial detector. The adversarial detector can be transferred to unknown attackers according to our experiments. Our experiments show that `Rapid` is robust in recognizing and defending against a wide range of unknown adversarial attacks, such as `Clare` (Li et al., 2021) and large language models like `ChatGPT-3.5` (OpenAI, 2023).

Finally, we have successfully developed a user-friendly application interface to serve as a challenging benchmark tool for evaluating the performance of adversarial attackers under the defense of `Rapid`. This is an important step to eliminate the evaluation variance among different codebases. We will release this tool after the review process because the codes are hard to anonymize.

## 2 PROPOSED METHOD

Our proposed framework comprises two phases. `Phase #1` is designed to train a joint model capable of performing both standard classification and adversarial detection. The adversarial detector's role is to identify adversaries and distinguish them from natural examples. `Phase #2` is dedicated to implementing pseudo-supervised adversary defense based on `PD`, to divert the victim model's attention from malicious perturbations and rectify the outputs without compromising performance on natural examples.

We will elaborate on the methodology of our framework step by step in the following subsections.

### 2.1 PHASE #1: ADVERSARIAL DETECTOR TRAINING

To implement an effective adversarial detector to pre-detect adversaries before the defense, we incorporate a binary classifier (a.k.a., in-victim-model adversarial detector) during the victim model training. We will show the adversarial detector's generalizability against unknown attackers in the experiment section.

**Multi-Attack-based Adversary Sampling**. This preprocessing step generates a set of adversaries to compose the dataset used for training the adversarial detector. In order to enable the adversarial detector to identify various unknown adversaries, we employ three adversarial attack methods: `BAE` (Garg & Ramakrishnan, 2020), `PWWS` (Ren et al., 2019), and `TextFooler` (Jin et al., 2020), to attack the victim classifier and sample adversaries, respectively. It's worth noting that we collect all adversaries, whether they were successful or failed attempts to deceive the victim model. In practice, given a dataset $\mathcal{D}$ consisting of natural examples and a victim classifier $F_S$ trained on $\mathcal{D}$, for all $(\mathbf{x}, y) \in \mathcal{D}$, we apply each of the adversarial attack methods to sample three adversaries as follows[1]:

$$(\tilde{\mathbf{x}}, \tilde{y})_i \leftarrow \mathcal{A}_i\left(F_S, (\mathbf{x}, y)\right), \tag{1}$$

where $\mathcal{A}_i$, $i \in \{1, 2, 3\}$, represents `BAE`, `PWWS`, and `TextFooler`, respectively. $(\tilde{\mathbf{x}}, \tilde{y})_i$ is the generated adversary. All the sampled adversaries and natural examples together constitute the adversarial dataset $\tilde{\mathcal{D}}$. We employ these three widely-used open-source adversarial attack methods for a proof-of-concept evaluation. Please note the defender in `Rapid` is decoupled with the adversarial detector and the sampling attackers do not influence the performance of the defender. The results in Table 4 indicate that the adversarial detector can adapt to unknown attack methods, even when trained on a small set of adversaries.

**Joint Model Training**. To improve the efficiency of adversary detection, we aim to train the adversarial detector concurrently with the victim model. We collect the adversaries $\tilde{\mathcal{D}}$ generated by multiple attackers and combine them with the natural (i.e., original) examples $\mathcal{D}$ to create a synthetic dataset $\overline{\mathcal{D}}$. In order to conduct joint model training, we propose three training objectives in the following subsection.

**Training Objectives**. The training objectives used for training the joint model $F_J$ include standard classification, adversarial training, and adversarial detection, respectively.

*Standard Classification Objective* ($\mathcal{L}_c$): Given that our experiments focus on text classification models, we employ the standard classification objective. To implement this objective, we denote the

---

[1]The formulation of word-level adversarial attack is available in Appendix A.

original labels as $\overline{y}_1$, $0 \leq \overline{y}_1 \leq C$, with $C$ representing the number of categories in the original dataset. It's important to note that only the prediction results of natural examples in $\overline{\mathcal{D}}$ are considered in the cross-entropy loss function. In other words, the $\overline{y}_1$ values for the adversaries in the synthetic dataset $\overline{\mathcal{D}}$ are set to a dummy value $\overline{y}_1 = \varnothing$ and are therefore ignored in this objective. The $\mathcal{L}_c$ is calculated as follows:

$$\mathcal{L}_c := -\sum_{i=1}^{C} \left[ p_i \log\left(\hat{p}_i\right) + q_i \log\left(\hat{q}_i\right) \right], \tag{2}$$

where $p$ and $\hat{p}$ represent the true and predicted probability distributions of the standard classification label (i.e., $\overline{y}_1$) of $\tilde{x}$. $q$ and $\hat{q}$ indicate any incorrect standard classification label and its likelihood.

*Adversarial Detection Objective* ($\mathcal{L}_d$): This objective is used to train the adversarial detector, which determines whether the input example is an adversary or not. The adversarial detector is a binary classifier that requires another independent label $\overline{y}_2$, where $\overline{y}_2 \in \{0, 1\}$. This objective only calculates the binary cross-entropy for both the natural examples and adversaries in $\overline{\mathcal{D}}$, where $\overline{y}_2$ is 0 and 1, respectively.

*Adversarial Training Objective* ($\mathcal{L}_a$): We also employ the adversarial training objective to enhance the robustness of adversaries. The difference, compared to existing adversarial training works, is that $\mathcal{L}_a$ uses an independent label $\overline{y}_3$, where $0 \leq \overline{y}_3 \leq C$. Please note that this objective only calculates the loss function for the adversaries. This means that $\overline{y}_3$ for natural examples is set to a dummy value $\overline{y}_1 = \varnothing$ and is therefore ignored in this objective. This approach prevents the adversarial training objective from negatively impacting the performance on pure natural examples and this problem are very common in recent works (Dong et al., 2021a;b). The calculation of $\mathcal{L}_a$ is the same as $\mathcal{L}_c$.

To accommodate these training objectives, each example $(\overline{x}, \overline{y}) \in \overline{\mathcal{D}}$ is augmented with three different labels for the three training objectives, i.e., $\overline{y} := (\overline{y}_1, \overline{y}_2, \overline{y}_3)^\top$.

**Overall Training Objective**. Finally, the three training objectives mentioned above are aggregated into a single loss function:

$$\mathcal{L} := \mathcal{L}_c + \mathcal{L}_d + \mathcal{L}_a + \lambda ||\boldsymbol{\theta}||_2^2, \tag{3}$$

where $\mathcal{L}_c$, $\mathcal{L}_d$ and $\mathcal{L}_a$ correspond to the losses for training a standard classifier, an adversarial detector, and adversarial training, respectively. $\lambda$ denotes the $L_2$ regularization parameter and $\boldsymbol{\theta}$ represents the parameters of the underlying PLM.

## 2.2 Phase #2: Reactive Adversarial Defense

In this section, we introduce a novel adversarial defense method. The adversarial defense in `Rapid` is designed to be independent of adversarial detectors and can accommodate future adversarial detection techniques. The defense method in `Rapid` includes the following components.

**Adversarial Defense Detection**. Prior approaches to adversarial defense often ignore pre-detect adversaries and run defense for all input texts, but defending against all inputs can be resource-intensive (Dong et al., 2021a;b). Therefore, we adopt a reactive adversarial defense mechanism to mitigate resource consumption. For instance, the joint model can determine whether the input example is adversarial using the following prediction:

$$(\hat{y}_1, \hat{y}_2, \hat{y}_3) \leftarrow F_J(\hat{x}), \tag{4}$$

where $\hat{x}$ represents the input example, and $\hat{y}_1$, $\hat{y}_2$, $\hat{y}_3$[2] are predictions based on the three training objectives, respectively. $F_J$ is the joint model trained in `Phase #1`. In `Rapid`, we only apply adversarial defense to inputs identified as adversaries (i.e., $\hat{y}_2 = 1$) by the joint model $F_J$. The repair of adversaries is conducted through perturbation defocusing.

**Perturbation Defocusing**. The goal of the `PD` defense is employing any adversarial attacker $\hat{\mathcal{A}}_{PD}$[3] to inject safe perturbations into the identified adversary $\hat{x}$ by adversarial attacks. In other words, we utilize an adversarial attacker to 'attack' the adversary, which actually repairs the semantics. In

---

[2]$\hat{y}_3$ is not used in `Phase #2`.

[3]We choose `PWWS` because it is cost-effective, and it can be replaced by any (or an ensemble of) adversarial attackers.

equation (4), we can obtain the pseudo label $\hat{y}_1$ of the adversary at an imperceptibly low cost. We use $\hat{y}_2$ to guide $\hat{\mathcal{A}}_{PD}$ in preventing repaired adversaries from retaining the same pseudo label. The defense process can be formulated as follows:

$$(\tilde{\mathbf{x}}^{\mathrm{r}}, \tilde{y}_1^{\mathrm{r}}) \leftarrow \hat{\mathcal{A}}_{PD}\left(F_J, (\hat{\mathbf{x}}, \hat{y}_1)\right), \tag{5}$$

where $(\tilde{\mathbf{x}}^{\mathrm{r}}, \tilde{y}_1^{\mathrm{r}})$ represents the repaired example and new prediction queried by $\hat{\mathcal{A}}_{PD}$ based on the victim model.

In `PD`, the perturbation introduced by the adversarial attacker is considered 'safe' since it does not alter the semantics of the adversary $\hat{\mathbf{x}}$. The rationale behind this perturbation is to divert the standard classifier's focus away from the malicious perturbations, allowing the standard classifier to concentrate on the adversary's original semantics. In essence, the repaired examples can be correctly classified based on their own robustness.

**Pseudo-similarity Supervision**. To prevent repaired adversaries from being misclassified, we propose a feature-level pseudo-semantic similarity filtering strategy to mitigate semantic bias `Rapid` generate a set of $k$ (i.e., we set $k = 3$ for efficiency) repaired examples for an adversary, we denote the set as $\mathcal{S} = \{\tilde{\mathbf{x}}_1^{\mathrm{r}}, \cdots, \tilde{\mathbf{x}}_i^{\mathrm{r}}, \cdots \tilde{\mathbf{x}}_k^{\mathrm{r}}\}$. We encode those repaired examples to extract the semantic features using $F_J$. We obtain the cosine similarity scores between $\tilde{\mathbf{x}}_i^{\mathrm{r}}$ and the rest repaired examples in $\mathcal{S}$ and calculate the average cosine similarity scores $s_i$ involving $\tilde{\mathbf{x}}_i^{\mathrm{r}}$ as follows:

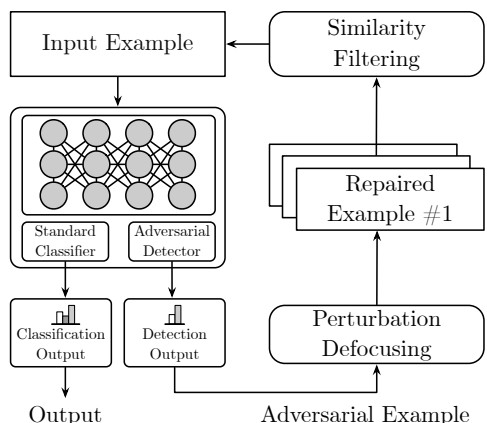

Figure 3: The visualization of the workflow of adversary defense in `Phase #2`.

$$s_i = \frac{\sum_{j=1}^{k} CosSim(H_i, H_j)}{k}, \tag{6}$$

where $H_i$ and $H_j$ are the hidden states encoded by the joint model $F_J$. $CosSim$ is the function for cosine similarity score calculation. After the defense, `Rapid` outputs the predicted label of the repaired example which has the largest average similarity score, i.e., $i$-th repaired example $\tilde{\mathbf{x}}_i^{\mathrm{r}}$ in $\mathcal{S}$, where $\forall j \in \{1, \cdots, k\}, s_i \geq s_j$.

## 3 EXPERIMENTAL SETTINGS

In this section, we introduce the experimental settings used in our experiments.

**Victim Models**. Any PLM can be used in a plug-in manner in our proposed framework. Without loss of generality, here we consider `BERT` (Devlin et al., 2019) and `DeBERTa` (He et al., 2021), two widely used PLMs from transformers[4], as both the victim classifier and the joint model. The corresponding hyperparameter settings of `BERT` and `DeBERTa` can be found in Appendix B.2.

Table 1: The statistics of datasets used for evaluating `Rapid`. We use subsets from **Amazon**, **AGNews** and **Yahoo!** datasets to evaluate `Rapid` as the previous works due to high resource occupation.

**Datasets**. Three widely-used text classification datasets[5] are considered in our experiments, including **SST2** (Socher et al., 2013), **Amazon** (Zhang et al., 2015), and **AGNews** (Zhang et al., 2015). Some of their statistics are listed in Table 1. **SST2** and **Amazon** are binary sentiment classification datasets. **AGNews** is a multi-categorical news classification dataset containing four categories. **Yahoo!** is another multi-categorical dataset that contains 10 categories.

| Dataset | Categories | Number of Examples | | |
|---|---|---|---|---|
| | | Training | Valid | Testing |
| SST2 | 2 | 6920 | 872 | 1821 |
| Amazon | 2 | 7000 | 1000 | 2000 |
| AGNews | 4 | 120000 | 0 | 7600 |
| Yahoo! | 10 | 1400000 | 0 | 60000 |

---

[4]https://github.com/huggingface/transformers
[5]We have released the detailed source codes and processed datasets in the supplementary materials.

**Adversarial Attackers**. Three open-source attackers provided by TextAttack[6] (Morris et al., 2020) are considered in our experiments while their working mechanisms are outlined in Appendix B.1. There are three roles of adversarial attackers are outlined as follows.

a) *Adversary Sampling*. `BAE`, `PWWS` and `TextFooler` are used to sample adversaries for training the adversarial detector. These attackers represent different types of attacks, thus enabling the training of a detector to be capable of recognizing a variety of adversarial attacks.

b) *Adversary Repair*. For perturbation defocusing, we employ `PWWS` as the attacker $\hat{\mathcal{A}}_{PD}$ in Section 2.2. Compared to open-source adversarial attackers such as `BAE`, `PWWS` rarely alters the semantics of natural examples in our observations and is slightly faster than other open-source attackers, such as `TextFooler`.

c) *Generalizability Evaluation*. We use `IGA` (Wang et al., 2021a), `DeepWordBug` (Gao et al., 2018), `PSO` (Zang et al., 2020) and `Clare` to evaulate `Rapid`'s generalizability.

**Evaluation Metrics**. In our experiments, we use the following five fine-grained metrics[7] for text classification to evaluate the adversarial defense performance.

**Nat. Acc.**: The natural accuracy is the victim's performance on the target dataset that only contains natural examples.

**Att. Acc.**: The accuracy under attacks denotes the victim's performance under adversarial attacks.

**Det. Acc.**: The detection accuracy measures the defender's performance of adversaries detection.

**Def. Acc.**: The defense accuracy denotes the defender's performance of adversaries repair.

**Rep. Acc.**: The paired accuracy evaluates the victim's performance on the attacked dataset after being repaired.

Unlike previous research (Xu et al., 2022; Yang et al., 2022; Dong et al., 2021a;b) that only evaluated a small amount of data extracted from the testing set, we evaluate the adversarial detection and defense performance on the entire testing set.

**Baseline Methods**. The performance of our proposed `Rapid` is compared against the following six adversarial defense baselines.

**DISP** (Zhou et al., 2019): It is an embedding feature reconstruction method for adversarial defense. `DISP` uses a perturbation discriminator to evaluate the probability that a token is perturbed and provides a set of potential perturbations. For each potential perturbation, an embedding estimator learns to restore the embedding of the original word based on the context.

**FGWS** (Mozes et al., 2021): It uses frequency-guided word substitutions to exploit the frequency properties of adversarial word substitutions for the detection of adversarial examples.

**RS&V** (Wang et al., 2022b): It is a text reconstruction method based on the randomized substitution-to-vote strategy. `RS&V` accumulates the logits of massive samples generated by randomly substituting the words in the adversaries with synonyms.

We cannot compare with some methods, e.g., `Textshield` (Shen et al., 2023) because there is no released source code or we cannot reproduce the experimental results. However, we can provide additional experimental results provided that there are any questions.

## 4 MAIN EXPERIMENTS

**Adversary Detection Performance**. Our experimental results, as shown in Table 2, demonstrate the effectiveness of the adversarial detector in `Rapid`. This in-victim-model adversarial detector, trained in conjunction with the standard classifier, accurately identifies adversaries across most datasets. Compared to the previous adversary detection-based defense (Mozes et al., 2021; Wang et al., 2022b; Shen et al., 2023), the in-victim-model adversarial detector identifies the adversaries with no extra cost. On the other hand, our evaluation confirms a very low false positive rate (approximately 2%) of adversary detection on natural examples, resulting in a very slight performance degradation on natural examples. Moreover, we showcase the adaptability of `Rapid` to previously unseen attack methods in Table 4, highlighting the versatility of our adversarial detector. It excels at identifying adversaries by detecting disruptions introduced by malicious attackers, such as grammar errors and word misuse. However, it's worth noting that detection performance on the **AGNews**

---

[6]`https://github.com/QData/TextAttack`
[7]The mathematical definitions of these evaluation metrics can be found in Appendix B.3.

Table 2: The main adversarial detection and defense performance of `Rapid` on four public datasets. The victim model is `BERT` and the results in **bold** font indicate the best performance. We report the average accuracy of five random runs. The adversarial defense performance reported in previous works varies from adversarial attackers' implementations. For fair comparisons, all the baseline experiments are re-implemented based on the latest adversarial attackers from the Textattack library to avoid biases. "TF" indicates `TextFooler`.

| Defender | Attacker | AGNews (4-category) | | | | | Yahoo! (10-category) | | | | | SST2 (2-category) | | | | | Amazon (2-category) | | | | |
|---|---|---|---|---|---|---|---|---|---|---|---|---|---|---|---|---|---|---|---|---|---|
| | | Nat. Acc. | Att. Acc. | Det. Acc. | Def. Acc. | Rep. Acc. | Nat. Acc. | Att. Acc. | Det. Acc. | Def. Acc. | Rep. Acc. | Nat. Acc. | Att. Acc. | Det. Acc. | Def. Acc. | Rep. Acc. | Nat. Acc. | Att. Acc. | Det. Acc. | Def. Acc. | Rep. Acc. |
| DISP | PWWS | | 32.09 | 55.49 | 57.82 | 68.23 | | 5.70 | 61.67 | 54.95 | 50.24 | | 23.44 | 38.93 | 34.46 | 35.33 | | 15.56 | 41.90 | 45.92 | 59.80 |
| | TF | 94.13 | 50.50 | 53.78 | 56.18 | 70.16 | 75.63 | 13.60 | 50.73 | 57.48 | 53.18 | 91.24 | 16.21 | 37.80 | 34.37 | 37.16 | 93.67 | 21.77 | 43.10 | 47.15 | 60.56 |
| | BAE | | 74.80 | 45.26 | 45.75 | 81.39 | | 27.50 | 54.82 | 53.75 | 50.90 | | | 35.21 | 36.59 | 37.51 | 42.22 | | 44.00 | 40.28 | 42.74 | 61.85 |
| FGWS | PWWS | | 32.09 | 65.24 | 68.35 | 71.78 | | 5.70 | 65.83 | 61.46 | 53.28 | | 23.44 | 40.28 | 40.38 | 39.20 | | 15.56 | 44.47 | 56.89 | 60.29 |
| | TF | 94.25 | 50.50 | 68.88 | 70.71 | 73.40 | 76.24 | 13.60 | 68.57 | 65.17 | 54.53 | 91.34 | 16.21 | 42.79 | 41.05 | 41.53 | 94.26 | 21.77 | 45.75 | 58.74 | 61.51 |
| | BAE | | 74.80 | 44.29 | 47.95 | 83.57 | | 27.50 | 58.63 | 56.33 | 52.94 | | | 35.21 | 43.83 | 48.37 | 44.90 | | 44.00 | 42.26 | 43.04 | 64.63 |
| RS&V | PWWS | | 32.09 | 83.67 | 84.96 | 83.80 | | 5.70 | 65.01 | 65.22 | 57.22 | | 23.44 | 36.90 | 37.10 | 38.54 | | 15.56 | 29.60 | 45.30 | 46.17 |
| | TF | 94.14 | 50.50 | 82.44 | 83.45 | 82.53 | 76.39 | 13.60 | 74.21 | 74.54 | 58.10 | 91.55 | 16.21 | 39.70 | 38.40 | 39.70 | 94.32 | 21.77 | 40.70 | 42.30 | 55.70 |
| | BAE | | 74.80 | 46.98 | 48.67 | 86.90 | | 27.50 | 37.41 | 37.88 | 62.27 | | | 35.21 | 19.84 | 20.92 | 43.65 | | 44.00 | 38.59 | 39.01 | 65.03 |
| Rapid | PWWS | | 32.09 | **90.11** | **95.88** | **92.36** | | 5.70 | **87.33** | **92.47** | **69.40** | | 23.44 | **94.03** | **98.62** | **89.85** | | 15.56 | **97.33** | **99.99** | **94.42** |
| | TF | 94.30 | 50.50 | **90.29** | **96.76** | **92.14** | 76.45 | 13.60 | **87.49** | **93.54** | **70.50** | 91.70 | 16.21 | **94.03** | **99.86** | **89.72** | 94.24 | 21.77 | **93.85** | **99.99** | **93.96** |
| | BAE | | 74.80 | **57.55** | **96.25** | **93.64** | | 27.50 | **82.46** | **96.30** | **73.06** | | | 35.21 | **78.99** | **99.28** | **89.77** | | 44.00 | **80.55** | **99.99** | **93.89** |

dataset is lower due to the absence of news data in the BERT training corpus, as referenced in He et al. (2021) (Table 8).

**Adversary Defense Performance**. In the realm of adversary defense, `Rapid` outperforms existing methods across all datasets, as outlined in Table 2. When we focus on correctly identified adversaries, `Rapid` can effectively repair up to 92-99% of them, even on the challenging 10-category Yahoo datasets. Our research also sheds light on the limitations of unsupervised text-level and feature-level reconstruction methods, exemplified in studies such as Zhou et al. (2019); Mozes et al. (2021); Wang et al. (2022b). These methods struggle to rectify the deep semantics in adversaries, rendering them inefficient and inferior. Additionally, we find that previous methods are not robust when defending against adversaries in short texts, as evidenced by their failure on the **SST2** and **Amazon** datasets. In contrast, `Rapid` consistently achieves higher defense accuracy, particularly on binary classification datasets.

In summary, `Rapid` employs adversarial attackers to repair adversaries' deep semantics and minimize edits in text space, resulting in promising adversarial defense performance. We emphasize the importance of dedicated deep semantics repair in the context of adversarial defense against unsupervised feature space and text space reconstruction.

**Ablation Experiment**. We conducted ablation experiments to assess the effectiveness of pseudo-similarity filtering. Pseudo-similarity filtering exclusively affects the defense process, so we have omitted unaffected metrics, such as detection accuracy, which can be found in Table 2. The experimental results are presented in Table 3. It is observed that the adversarial defense performance of `Rapid` without similarity filtering is notably inferior (approximately 1%) in most scenarios. Furthermore, the degradation in defense performance is more pronounced in the case of the **AGNews** and **Yahoo!** datasets compared to the **SST2** and **Amazon** datasets. This discrepancy is attributed to the larger vocabularies and longer text lengths in the **AGNews** and **Yahoo!** datasets, resulting in diversified repaired examples in terms of similarity.

Table 3: The performance of `Rapid` **without pseudo-similarity filtering**, with colored numbers indicating performance declines in the ablated `Rapid`. We omit the metrics that are not unaffected by pseudo-similarity filtering.

| Dataset | Attacker | Def. Acc. | Rep. Acc. |
|---|---|---|---|
| AGNews | PWWS | 94.19 (- 1.69) | 90.80 (- 1.56) |
| | TF | 94.26 (- 2.50) | 91.35 (- 0.79) |
| | BAE | 92.98 (- 3.27) | 91.44 (- 2.20) |
| Yahoo! | PWWS | 88.04 (- 4.43) | 65.38 (- 4.02) |
| | TF | 91.28 (- 2.26) | 67.48 (- 3.02) |
| | BAE | 92.48 (- 3.84) | 71.35 (- 1.71) |
| SST2 | PWWS | 98.12 (- 0.50) | 87.80 (- 2.05) |
| | TF | 98.03 (- 1.83) | 88.40 (- 1.32) |
| | BAE | 95.87 (- 3.41) | 87.52 (- 2.25) |
| Amazon | PWWS | 99.99 ( 0.00) | 94.40 (- 0.02) |
| | TF | 98.92 (- 1.07) | 93.31 (- 0.65) |
| | BAE | 98.53 (- 1.41) | 93.62 (- 0.27) |

## 4.1 RESEARCH QUESTIONS

We discuss more findings about `Rapid` by answering the following research questions (RQs).

**RQ1: How is the generalization ability of `Rapid` to unknown attackers?**

_Methods_: To extensively assess the generalization ability of the in-victim-model adversarial detector in `Rapid`, we have conducted experiments among various state-of-the-art adversarial attackers: `PSO`, `IGA`, `DeepWordBug`, and `Clare`. These adversarial attackers were not included in the training of the adversarial detector in `Rapid`. We hope our experiments can attract attention to the generalization ability of adversarial detectors. It's important to note that better adversarial detection and defense performance against unknown adversarial attackers indicates the superior generalizability of `Rapid`.

_Results_: From the statistical comparison results presented in Table 4, we observe that `Rapid` can identify up to 98.67% of adversaries on both the **SST2** and Amazon datasets when considering adversarial detection performance. In terms of adversarial defense, `Rapid` is capable of repairing a substantial number of adversaries generated by various unknown attack methods (up to 87.68% and 94.65% on the **SST2** and **Amazon** datasets, respectively). However, `Rapid` experiences a decline in performance in identifying and defending against adversaries when facing the challenging `Clare` attack. This drop in performance is likely attributable to the subpar accuracy of adversarial detection, which could potentially be improved by training `Clare`-based adversaries for adversarial detection within `Rapid`. In summary, `Rapid` has demonstrated robust generalization ability, effectively detecting and repairing a wide array of adversaries generated by unknown attackers.

Table 4: The performance of `Rapid` for adversarial detection and defense **against unknown adversarial attacks**.

| Dataset | Attacker | Att. Acc. | Det. Acc. | Def. Acc. | Rep. Acc. |
|---|---|---|---|---|---|
| AGNews | PSO | 14.83 | 68.46 | 67.82 | 90.39 |
| | IGA | 26.87 | 76.74 | 74.59 | 92.33 |
| | DeepWordBug | 45.53 | 72.73 | 87.23 | 89.33 |
| | Clare | 8.46 | 62.78 | 61.54 | 64.78 |
| Yahoo! | PSO | 6.28 | 80.26 | 76.89 | 87.82 |
| | IGA | 14.75 | 82.69 | 81.02 | 54.55 |
| | DeepWordBug | 51.34 | 72.73 | 87.10 | 62.27 |
| | Clare | 3.56 | 64.85 | 62.40 | 52.47 |
| SST2 | PSO | 7.95 | 87.50 | 87.50 | 82.61 |
| | IGA | 18.39 | 89.33 | 98.67 | 87.68 |
| | DeepWordBug | 30.67 | 95.44 | 83.59 | 81.90 |
| | Clare | 2.59 | 62.50 | 59.37 | 65.30 |
| Amazon | PSO | 5.76 | 90.48 | 90.48 | 91.55 |
| | IGA | 14.91 | 92.31 | 92.31 | 94.65 |
| | DeepWordBug | 43.43 | 87.04 | 85.19 | 86.87 |
| | Clare | 3.25 | 60.44 | 59.37 | 62.94 |

**RQ2: Does perturbation defocusing really repair adversaries?**

_Methods_: To address this question, we investigate the discrepancy between adversaries and their repaired counterparts in the feature space. Specifically, we employ three attackers (i.e., `BAE`, `PWWS`, `TextFooler`) to generate adversaries and their corresponding repaired examples, considering a random selection of 1000 natural examples. Using the victim model, we encode these examples into the feature space and evaluate the cosine similarity between adversary-natural example pairs and repaired adversary-natural example pairs. The larger cosine similarity scores indicate better performance in repairing the deep semantics in the adversaries.

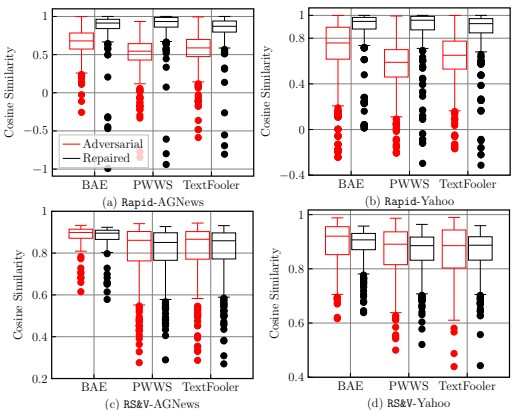

Figure 4: The box plots of semantic cosine similarity score distributions on multi-categorial datasets. Similar to Figure 1, `Rapid` is more competent to repair semantics according to the feature similarity score distributions.

_Results_: Figure 1 and Figure 4 depict box plots illustrating the similarity score distributions collected from pairwise semantic similarity assessments. The semantic similarity score distributions (e.g., the median similarity scores of repaired examples are always larger than the adversaries) from these plots reveal a notable global similarity between the natural examples and repaired examples by `Rapid`, which means `Rapid` does repair the deep semantics of the adversaries. Conversely, it is apparent that the similarity scores of the repaired examples obtained using `RS&V` are indistinguishable from the adversarial examples across all datasets. This situation happens to many of the existing adversarial defense methods. In conclusion, our observations show the ability of `Rapid` to effectively repair the deep semantics of adversaries.

**RQ3: How does the inherent robustness of the victim model affect `Rapid`?**

*Methods*: In this research question, we assessed the impact of the inherent robustness of the victim model, focusing on `DeBERTa`, a cutting-edge PLM utilized across various tasks. Specifically, we trained a victim model based on `DeBERTa`, replicating the experimental setup and evaluating the performance variation of `Rapid` based on this `DeBERTa` victim model.

*Results*: The outcomes are elaborated in Table 5. When compared to the victim model built on `BERT`, the `DeBERTa`-based victim model demonstrates superior accuracy under adversarial attacks, indicating higher inherent robustness in `DeBERTa` compared to `BERT`. We observed that our method based on `DeBERTa` generally excels in accurately identifying adversaries across all classification datasets, especially on the binary datasets, in comparison to `BERT`. The performance in adversarial detection and defense fol-

Table 5: The performance of `Rapid` on four public datasets **based on the victim model `DeBERTa`**. The numbers in red color indicate performance declines compared to the `BERT`-based `Rapid`.

| Dataset | Attacker | Nat. Acc. | Att. Acc. | Det. Acc. | Def. Acc. | Rep. Acc. |
|---------|----------|-----------|-----------|-----------|-----------|-----------|
| **AGNews** | PWWS | 96.69 | 62.77 | 96.47 | 98.47 | 93.12 |
|  | TF |  | 39.85 | 91.41 | 95.90 | 93.69 |
|  | BAE |  | 81.64 | 90.20 | 97.92 | 93.40 |
| **Yahoo!** | PWWS | 78.63 | 15.70 | 88.91 | 92.64 | 70.47 |
|  | TF |  | 6.19 | 89.32 | 92.60 | 69.96 |
|  | BAE |  | 47.50 | 90.25 | 93.74 | 72.12 |
| **SST2** | PWWS | 95.01 | 37.14 | 95.21 | 98.42 | 94.15 |
|  | TF |  | 22.59 | 93.06 | 99.08 | 94.58 |
|  | BAE |  | 38.84 | 80.82 | 98.59 | 94.16 |
| **Amazon** | PWWS | 95.51 | 22.72 | 97.62 | 99.99 | 94.55 |
|  | TF |  | 23.95 | 94.91 | 99.99 | 94.84 |
|  | BAE |  | 56.65 | 82.71 | 99.99 | 94.50 |

lows a similar upward trajectory. Emphasizing the substantial influence of the victim model's robustness on our method, particularly in enhancing adversarial detection and defense.

## 5 RELATED WORKS

Prior research on adversarial defense can be classified into three categories: adversarial training-based methods (Miyato et al., 2017; Zhu et al., 2020; Ivgi & Berant, 2021); context reconstruction-based methods (Pruthi et al., 2019; Liu et al., 2020b; Mozes et al., 2021; Keller et al., 2021; Chen et al., 2021; Xu et al., 2022; Li et al., 2022; Swenor & Kalita, 2022); and feature reconstruction-based methods(Zhou et al., 2019; Jones et al., 2020; Wang et al., 2021a). Some studies (Wang et al., 2021b) also investigated hybrid defense methods. As for the adversarial training-based methods, they are notorious for the performance degradation of natural examples. They can improve the robustness of PLMs by fine-tuning, yet increasing the cost of model training caused by catastrophic forgetting (Dong et al., 2021b). Text reconstruction-based methods, such as word substitution (Mozes et al., 2021; Bao et al., 2021) and translation-based reconstruction, may fail to identify semantically repaired adversaries or introduce new malicious perturbations (Swenor & Kalita, 2022). Feature reconstruction methods, on the other hand, may struggle to repair typo attacks (Liu et al., 2020a; Tan et al., 2020; Jones et al., 2020), sentence-level attacks (Zhao et al., 2018; Cheng et al., 2019), and other unknown attacks. There are some works towards the adversarial detection and defense joint task(Zhou et al., 2019; Mozes et al., 2021; Wang et al., 2022b). However, these adversarial detection methods may be ineffective for unknown adversarial attackers and can hardly alleviate resource waste in adversarial defense. Another similar work to `Rapid` is `Textshield` (Shen et al., 2023), which aims to defend against word-level adversarial attacks by detecting adversarial sentences based on a saliency-based detector and fixing the adversarial examples using a corrector. Overall, our study focuses on maintaining the semantics by introducing minimal safe perturbations into adversaries, thus alleviating the semantic shifting problem in all reconstruction-based works.

## 6 CONCLUSION

We propose a novel adversarial defense method, i.e., perturbation defocusing, to defend against adversarial examples. Our method almost addresses the semantic shifting problem in the previous studies. In our experiments, `Rapid` shows an impressive performance in repairing adversarial examples (up to $\sim 99\%$ of correctly identified adversarial examples). We argue that perturbation defocusing has the potential to significantly shift the landscape of textual adversarial defense. While further research is needed to fully explore the potential of perturbation defocusing, it is clear that it holds promise for improving the accuracy and robustness of adversarial defense in the future.

## 7 REPRODUCIBILITY

To encourage everyone interested in our work to implement `Rapid`, we have taken the following steps:

- We have created an online click-to-run demo alailable at `https://tinyurl.com/22ercuf8` for easy evaluation. Everyone can input adversarial examples and obtain the repaired examples immediately.
- We have released the detailed source codes and processed datasets that can be retrieved in the supplementary materials. This enables everyone to access the official implementation, aiding in understanding the paper and facilitating their own implementations.
- We will also release an online benchmark tool for evaluating the performance of adversarial attackers under the defense of `Rapid`. This step is essential for reducing evaluation variance across different codebases.

These efforts are aimed at promoting the reproducibility of our work and facilitating its implementation by the research community.

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

# A ADVERSARIAL ATTACK

## A.1 WORD-LEVEL ADVERSARIAL ATTACK

Our focus is on defending against word-level adversarial attacks. However, our method can be easily adapted to different types of adversarial attacks. Let $x = (x_1, x_2, \cdots, x_n)$ be a natural sentence, where $x_i$, $1 \leq i \leq n$, denotes a word. $y$ is the ground truth label. Word-level attackers generally replace some original words with similar words (e.g., synonyms) to fool the objective model. For example, substituting $x_i$ with $\hat{x}_i$ generates an adversary: $\hat{x} = (x_1, \cdots, \hat{x}_i, \cdots, x_n)$, where $\hat{x}_i$ is an alternative substitution for $x_i$. For an adversary $\hat{x}$, the objective model $F$ predicts its label as follows:

$$\hat{y} = \operatorname{argmax} F\left(\cdot | \hat{x}\right), \tag{7}$$

where $\hat{y} \neq y$ if $\hat{x}$ is a successful adversary. To represent adversarial attacks to $F$ using an adversarial attacker $\mathcal{A}$, we denote an adversarial attack as follows:

$$(\hat{x}, \hat{y}) \leftarrow \mathcal{A}(F, (x, y)), \tag{8}$$

where $x$ and $y$ denote the natural example and its true label. $\hat{x}$ and $\hat{y}$ are the perturbed adversary and label, respectively.

## A.2 INVESTIGATION OF TEXTUAL ADVERSARIAL ATTACK

This section delves into an examination of textual adversarial attacks.

Traditional approaches, such as those noted by Li et al. (2019) and Ebrahimi et al. (2018), often involve character-level modifications to words (e.g., changing "good" to "go0d") to deceive models by altering their statistical patterns. In a different approach, knowledge-based perturbations, exemplified by the work of Zang et al. (2020), employ resources like HowNet to confine the search space, especially in terms of substituting words.

Recent research (Garg & Ramakrishnan, 2020; Li et al., 2020) has investigated using pre-trained models for generating context-aware perturbations (Li et al., 2021). Semantic-based methods, such as SemAttack (Wang et al., 2022a), typically use BERT embedding clusters to create sophisticated adversarial examples. This differs from prior heuristic methods that employed greedy algorithms (Yang et al., 2020; Jin et al., 2020) or genetic algorithms (Alzantot et al., 2018; Zang et al., 2020), as well as gradient-based techniques (Wang et al., 2020; Guo et al., 2021) that concentrated on syntactic limitations.

With the evolution of adversarial attack techniques, numerous tools such as TextAttack (Morris et al., 2020) and OpenAttack (Zeng et al., 2021) have been developed and made available in the open-source community. These resources facilitate deep learning researchers to efficiently assess adversarial robustness with minimal coding. Therefore, our experiments in adversarial defense are conducted using the TextAttack framework, and we extend our gratitude to the authors and contributors of TextAttack for their significant efforts.

# B  EXPERIMENTS IMPLEMENTATION

## B.1  EXPERIMENTAL ADVERSARIAL ATTACKERS

We employ `BAE`, `PWWS`, and `TextFooler` to generate adversaries for training the adversarial detector. These attackers are chosen because they represent different types of attacks, allowing us to train a detector capable of recognizing a variety of adversarial attacks. This detector exhibits good generalization ability, which we confirm through experiments with other adversarial attackers such as `IGA`, `DeepWordBug`, `PSO`, and `Clare`. Including a larger number of adversarial attackers in the training process can further enhance the performance of the detector. We provide a brief introduction to these adversarial attackers:

a) **BAE** (Garg & Ramakrishnan, 2020) generates perturbations by replacing and inserting tagged words based on the candidate words generated by the masked language model (MLM). To identify the most important words in the text, `BAE` employs a word deletion-based importance evaluation method.

b) **PWWS** (Ren et al., 2019) is an adversarial attacker based on synonym replacement, which combines word significance and classification probability for word replacement.

c) **TextFooler** (Jin et al., 2020) considers additional constraints (such as prediction consistency, semantic similarity, and fluency) when generating adversaries. `TextFooler` uses a gradient-based word importance measure to locate and perturb important words.

## B.2  HYPERPARAMETER SETTINGS

We employ the following configurations for fine-tuning classifiers:

1. The learning rates for both `BERT` and `DeBERTa` are set to $2 \times 10^{-5}$.
2. The batch size is 16, and the maximum sequence modeling length is 128.
3. Dropouts are set to 0.1 for all models.
4. The loss functions of all objectives use cross-entropy.
5. The victim models and `Rapid` models are trained for 5 epochs.
6. The optimizer used for fine-tuning objective models is `AdamW`.

Please refer to our released code for more details.

## B.3  EVALUATION METRICS

In this section, we introduce the adversarial defense metrics. First, we select a target dataset, referred to as $\mathcal{D}$, containing only natural examples. Our goal is to generate adversaries that can deceive a victim model $F_J$. We group the successful adversaries into a subset called $\mathcal{D}_{adv}$ and the remaining natural examples with no adversaries into another subset called $\mathcal{D}_{nat}$. We then combine these two subsets to form the attacked dataset, $\mathcal{D}_{att}$. We apply `Rapid` to $\mathcal{D}_{att}$ to obtain the repaired dataset, $\mathcal{D}_{rep}$. The evaluation metrics used in the experiments are described as follows:

$$\text{Nat. Acc.} = \frac{TP_{\mathcal{D}} + TN_{\mathcal{D}}}{P_{\mathcal{D}} + N_{\mathcal{D}}}$$

$$\text{Att. Acc.} = \frac{TP_{\mathcal{D}_{att}} + TN_{\mathcal{D}_{att}}}{P_{\mathcal{D}_{att}} + N_{\mathcal{D}_{att}}}$$

$$\text{Det. Acc.} = \frac{TP^*_{\mathcal{D}_{adv}} + TN^*_{\mathcal{D}_{adv}}}{P^*_{\mathcal{D}_{adv}} + N^*_{\mathcal{D}_{adv}}}$$

$$\text{Def. Acc.} = \frac{TP_{\mathcal{D}_{adv}} + TN_{\mathcal{D}_{adv}}}{P_{\mathcal{D}_{adv}} + N_{\mathcal{D}_{adv}}}$$

$$\text{Rep. Acc.} = \frac{TP_{\mathcal{D}_{rep}} + TN_{\mathcal{D}_{rep}}}{P_{\mathcal{D}_{rep}} + N_{\mathcal{D}_{rep}}}$$

where $TP$, $TN$, $P$ and $N$ are the number of true positives and true negatives, positive and negative in standard classification, respectively. $TP^*$, $TN^*$, $P^*$ and $N^*$ indicate the case numbers in adversarial detection.

## B.4 EXPERIMENTAL ENVIRONMENT

The experiments are carried out on a computer running the Cent OS 7 operating system, equipped with an RTX 3090 GPU and a Core i-12900k processor. We use the PyTorch 1.12 library and a modified version of TextAttack, based on version 0.3.7.

## C ABLATION EXPERIMENTS

### C.1 DEFENSE OF LLM-BASED ADVERSARIAL ATTACK

Recent years have witnessed the superpower of large language models (LLMs) such as `ChatGPT` (OpenAI, 2023), which we hypothesize to have a stronger ability to generate adversaries. In this subsection, we evaluate the defense performance of `Rapid` against adversaries generated by `ChatGPT-3.5`. Specifically, for each dataset considered in our previous experiments, we use `ChatGPT`[8] to generate 100 adversaries and investigate the defense accuracy achieved by `Rapid`.

Table 6: Defense performance of `Rapid` against adversarial attacks generated by `ChatGPT-3.5`.

| Dataset | Attacker | Def. Acc. (%) | Rep. Acc. (%) |
|---------|----------|---------------|---------------|
| **AGNews** | ChatGPT | RS&V | 59.0 |
|            |          | Rapid | **72.0** |
| **Yahoo!** | ChatGPT | RS&V | 49.0 |
|            |          | Rapid | **61.0** |
| **SST2** | ChatGPT | RS&V | 37.0 |
|          |          | Rapid | **74.0** |
| **Amazon** | ChatGPT | RS&V | 58.0 |
|            |          | Rapid | **82.0** |

From the experimental results shown in Table 6, we find that `Rapid` consistently outperforms `RS&V` in terms of defense accuracy. Specifically, in the **SST2** dataset, `RS&V` records a defense accuracy of 37.0%, however, `Rapid` impressively repairs 74.0% of the attacks. Similar trends hold for the **Amazon** and **AGNews** datasets, where `Rapid` achieves defense accuracy of 82.0% and 72.0% respectively, in contrast to the 58.0% and 59.0% offered by `RS&V`. In conclusion, `Rapid` can defend against various unknown adversarial attacks which have a remarkable performance in contrast to existing adversarial defense approaches.

### C.2 PERFORMANCE OF RAPID BASED ON DIFFERENT $\hat{\mathcal{A}}_{PD}$

In `Rapid`, `PD` can incorporate any adversarial attacker or even an ensemble of attackers, as the process doesn't require prior knowledge of the specific malicious perturbations. Regardless of which adversaries are deployed against `Rapid`, `PWWS` consistently seeks safe perturbations for the current adversarial examples. The abstract nature of `PD` is critical, allowing for adaptability and effectiveness against a broad spectrum of adversarial attacks, rendering it a versatile defense mechanism in our study.

In order to investigate the impact of $\hat{\mathcal{A}}_{PD}$ in `Phase #2`, we have implemented further experiments to demonstrate the adversarial defense performance of `PD` using different attackers, e.g., `TextFooler` and `BAE`. The results are shown in Table 7. According to the experimental results, it is observed that `PWWS` has a similar performance to `TextFooler` in `PD`, while `BAE` is slightly inferior to both `PWWS` and `TextFooler`. However, the variance are not significant among different attackers in `PD`, which means the performance of `Rapid` is not sensitive to the choice of $\hat{\mathcal{A}}_{PD}$, in contrast to the adversarial attack performance of the adversarial attacker.

### C.3 PERFORMANCE OF RAPID WITHOUT ADVERSARIAL TRAINING OBJECTIVE

The rationale behind the adversarial training objective $\mathcal{L}_a$ in our study is founded on two key hypotheses.

a) **Enhancing Adversarial Detection:** We recognize an implicit link between the tasks of adversarial training and adversarial example detection. Our theory suggests that by incorporating an adversarial training objective, we can indirectly heighten the model's sensitivity to adversarial examples, leading to more accurate detection of such instances.

---

[8]`ChatGPT3.5-0301`

Table 7: The adversarial detection and defense performance of `Rapid` based on different backends ($\hat{\mathcal{A}}_{PD}$). We report the average accuracy of five random runs. "TF" indicates `TextFooler`.

| Defender | Attacker | AGNews (4-category) | | | | | Yahoo! (10-category) | | | | | SST2 (2-category) | | | | | Amazon (2-category) | | | | |
|---|---|---|---|---|---|---|---|---|---|---|---|---|---|---|---|---|---|---|---|---|---|
| | | Nat. Acc. | Att. Acc. | Det. Acc. | Def. Acc. | Rep. Acc. | Nat. Acc. | Att. Acc. | Det. Acc. | Def. Acc. | Rep. Acc. | Nat. Acc. | Att. Acc. | Det. Acc. | Def. Acc. | Rep. Acc. | Nat. Acc. | Att. Acc. | Det. Acc. | Def. Acc. | Rep. Acc. |
| Rapid(PWWS) | PWWS | | 32.09 | 90.11 | 95.88 | 92.36 | | 5.70 | 87.33 | 92.47 | 69.40 | | 23.44 | 94.03 | 98.62 | 89.85 | | 15.56 | 97.33 | 99.99 | 94.42 |
| | TF | 94.30 | 50.50 | 90.29 | 96.76 | 92.14 | 76.45 | 13.60 | 87.49 | 93.54 | 70.50 | 91.55 | 16.21 | 94.03 | 99.86 | 89.72 | 94.32 | 21.77 | 93.85 | 99.99 | 93.96 |
| | BAE | | 74.80 | 57.55 | 96.25 | 93.64 | | 27.50 | 82.46 | 96.30 | 73.06 | | 35.21 | 78.99 | 99.28 | 89.77 | | 44.00 | 80.55 | 99.99 | 93.89 |
| Rapid(TF) | PWWS | | 32.09 | 83.67 | 94.07 | 92.27 | | 5.70 | 65.01 | 83.25 | 65.33 | | 23.44 | 36.90 | 98.90 | 90.67 | | 15.56 | 29.60 | 99.99 | 94.33 |
| | TF | 94.30 | 50.50 | 82.44 | 96.46 | 92.67 | 76.45 | 13.60 | 74.21 | 92.96 | 71.00 | 91.55 | 16.21 | 39.70 | 99.98 | 90.73 | 94.32 | 21.77 | 40.70 | 99.99 | 94.33 |
| | BAE | | 74.80 | 46.98 | 92.68 | 91.00 | | 27.50 | 37.41 | 86.49 | 72.67 | | 35.21 | 19.84 | 99.98 | 91.33 | | 44.00 | 38.59 | 99.99 | 94.33 |
| Rapid(BAE) | PWWS | | 32.09 | 83.67 | 93.22 | 92.08 | | 5.70 | 65.01 | 81.15 | 64.00 | | 23.44 | 36.90 | 93.92 | 87.67 | | 15.56 | 29.60 | 99.54 | 94.00 |
| | TF | 94.30 | 50.50 | 82.44 | 95.96 | 92.33 | 76.45 | 13.60 | 74.21 | 87.79 | 67.33 | 91.55 | 16.21 | 39.70 | 96.55 | 89.00 | 94.32 | 21.77 | 40.70 | 99.61 | 93.64 |
| | BAE | | 74.80 | 46.98 | 95.12 | 91.33 | | 27.50 | 37.41 | 83.78 | 72.00 | | 35.21 | 19.84 | 97.55 | 90.00 | | 44.00 | 38.59 | 99.15 | 93.80 |

b) **Improving Model Robustness:** We posit that an adversarial training objective can bolster the model's robustness, thereby mitigating performance degradation when the model faces an attack. This approach is designed to strengthen the model against potential adversarial threats.

To validate these hypotheses, we conducted ablation experiments on the adversarial training objective. The experimental setup was aligned with that described in Table 2, and the results are outlined in Table 8.

Table 8: The adversarial detection and defense performance of `Rapid` with ("w/") and without ("w/o") the adversarial training objective. We report the average accuracy of five random runs. "TF" indicates `TextFooler`.

| Defender | Attacker | AGNews (4-category) | | | | | Yahoo! (10-category) | | | | | SST2 (2-category) | | | | | Amazon (2-category) | | | | |
|---|---|---|---|---|---|---|---|---|---|---|---|---|---|---|---|---|---|---|---|---|---|
| | | Nat. Acc. | Att. Acc. | Det. Acc. | Def. Acc. | Rep. Acc. | Nat. Acc. | Att. Acc. | Det. Acc. | Def. Acc. | Rep. Acc. | Nat. Acc. | Att. Acc. | Det. Acc. | Def. Acc. | Rep. Acc. | Nat. Acc. | Att. Acc. | Det. Acc. | Def. Acc. | Rep. Acc. |
| Rapid (w/ $\mathcal{L}_a$) | PWWS | | 32.09 | 90.11 | 95.88 | 92.36 | | 5.70 | 87.33 | 92.47 | 69.40 | | 23.44 | 94.03 | 98.62 | 89.85 | | 15.56 | 97.33 | 99.99 | 94.42 |
| | TF | 94.30 | 50.50 | 90.29 | 96.76 | 92.14 | 76.45 | 13.60 | 87.49 | 93.54 | 70.50 | 91.55 | 16.21 | 94.03 | 99.86 | 89.72 | 94.32 | 21.77 | 93.85 | 99.99 | 93.96 |
| | BAE | | 74.80 | 57.55 | 96.25 | 93.64 | | 27.50 | 82.46 | 96.30 | 73.06 | | 35.21 | 78.99 | 99.28 | 89.77 | | 44.00 | 80.55 | 99.99 | 93.89 |
| Rapid (w/o $\mathcal{L}_a$) | PWWS | | 11.10 | 82.88 | 92.07 | 90.70 | | 3.46 | 78.43 | 87.42 | 63.79 | | 10.70 | 91.41 | 99.62 | 89.60 | | 16.5 | 96.50 | 99.30 | 93.60 |
| | TF | 94.44 | 16.09 | 84.88 | 93.07 | 87.28 | 76.32 | 0.42 | 78.65 | 78.36 | 56.72 | 91.54 | 5.30 | 89.48 | 95.15 | 85.80 | 94.29 | 17.53 | 98.63 | 99.17 | 92.78 |
| | BAE | | 67.93 | 83.17 | 91.49 | 91.15 | | 45.10 | 71.89 | 75.47 | 64.56 | | 25.70 | 57.01 | 95.64 | 87.10 | | 45.54 | 92.67 | 99.48 | 93.31 |

These experimental findings reveal that omitting the adversarial training objective in `Rapid` consistently leads to a reduction in model robustness across all datasets. This reduction can be as substantial as approximately 30%, adversely affecting the performance of the adversarial defense. Additionally, adversarial detection capabilities also diminish, with the most significant drop being around 20%. These results highlight the critical role of the adversarial training objective in `Rapid`, confirming its efficacy in enhancing both model robustness and adversarial example detection capabilities.

## C.4 Performance of Rapid without Multitask Training Objective

Before developing `Rapid`, we carefully considered the potential impact on classification performance due to multitask training objectives. This consideration was explored in our proof-of-concept experiments.

To delve deeper into this impact, we trained victim models as single-task models (i.e., no adversarial detection objective and adversarial training objective), instead of multitask training, and then collated detailed results for comparison with `Rapid`. In this experiment, we focused solely on evaluating performance using pure natural examples. The results of this comparison are outlined in Table 9. The symbols "↑" and "↓" accompanying the numbers indicate whether the performance is better or worse than that of the single-task model, respectively.

Based on these results, it is apparent that the inclusion of additional loss terms in multitask training objectives does impact the victim model's performance on clean examples. However, this influence is not substantial across all datasets and shows only slight variations. This finding suggests that the impact of multitask training objectives is relatively minor when compared to traditional adversarial training methods.

| Dataset | Model | Victim-S | Victim-M |
|---------|-------|----------|----------|
| **AGNews** | BERT | 94.30 | 93.90 (−0.40 ↓) |
| **Yahoo!** | BERT | 76.45 | 76.61 (+0.16 ↑) |
| **SST2** | BERT | 91.70 | 91.49 (−0.21 ↓) |
| **Amazon** | BERT | 94.24 | 94.24 (—) |

Table 9: Victim model's accuracy (%) on clean dataset based single-task and multitask training scenarios, i.e., **Victim-S** and **Victim-M** respectively. The experiments are based on the BERT model.

## C.5 PERFORMANCE COMPARISON BETWEEN RAPID AND ADVERSARIAL TRAINING BASELINE

We have conducted experiments to showcase the experimental results of the adversarial training baseline (AT). The victim model is BERT, and the experimental setup is the same as for Rapid, including the number of adversaries used for training. We only show the metric of repaired accuracy, as AT does not support detect-to-defense. The results (i.e., Rep. Acc. (%)) are available in Table 10.

| Dataset | Attacker | Rapid | AT |
|---------|----------|-------|-----|
| **AGNews** | PWWS | 92.36 | 60.10 |
| | TF | 92.14 | 61.87 |
| | BAE | 93.64 | 63.62 |
| **Yahoo!** | PWWS | 69.40 | 40.21 |
| | TF | 70.50 | 38.75 |
| | BAE | 73.06 | 42.97 |
| **SST2** | PWWS | 89.85 | 32.46 |
| | TF | 89.72 | 31.23 |
| | BAE | 89.77 | 34.61 |
| **Amazon** | PWWS | 94.42 | 51.90 |
| | TF | 93.96 | 49.49 |
| | BAE | 93.89 | 49.75 |

Table 10: The repaired performance of Rapid and the adversarial training baseline. We report the average accuracy of five random runs. "TF" indicates TextFooler.

For these experiments, we used BERT as the victim model and maintained the same experimental setup as for Rapid, including the number of adversaries used for training. It's important to note that we focus solely on the repaired accuracy metric, as AT does not facilitate detect-to-defense functionality. From these results, it becomes apparent that the traditional adversarial training baseline is less effective compared to Rapid, which utilizes perturbation defocusing. Specifically, the adversarial defense accuracy of AT is generally below 50%. This finding underscores the limitations of traditional adversarial training methods, particularly their high cost and reduced effectiveness against adapted adversarial attacks.

## C.6 EFFICIENCY EVALUATION OF RAPID

The main efficiency depends on multiple adversarial perturbations search. We have implemented two experiments to investigate the efficiency of Rapid. Please note that the time costs for adversarial attack and defense are dependent on specific software and hardware environments.

**Time Costs for Multiple Examples**. We have collected three small sub-datasets that contain different numbers of adversarial examples and natural examples, say 200:0, 100:100, and 0:200. We apply adversarial detection and defense to this dataset and calculate the time costs. The results (measurement: second) are available in Table 11.

**Time Costs for Multiple Examples**. We have also detailed the time costs per natural example, adversarial attack, and adversarial defense in PD. The experimental results can be found in Table 12.

According to the experimental results, PD is slightly faster than the adversarial attack in most cases. Intuitively, the perturbed semantics in a malicious adversarial example are generally not robust, as

| Attacker | AGNews | | | Yahoo! | | | SST2 | | | Amazon | | |
|---|---|---|---|---|---|---|---|---|---|---|---|---|
| | 200:0 | 100:100 | 0:200 | 200:0 | 100:100 | 0:200 | 200:0 | 100:100 | 0:200 | 200:0 | 100:100 | 0:200 |
| PWWS | | 142.090 | 298.603 | | 313.317 | 621.196 | | 36.268 | 126.054 | | 438.532 | 875.083 |
| TF | 1.188 | 146.654 | 293.542 | 1.157 | 314.926 | 642.206 | 1.092 | 51.303 | 137.795 | 1.138 | 329.075 | 665.052 |
| BAE | | 141.434 | 260.231 | | 352.186 | 876.606 | | 52.626 | 138.325 | | 349.256 | 655.264 |

Table 11: The efficiency of `Rapid` defending against different adversarial attacks with different portions of natural and adversarial instances. The measurement is second.

| Defender | Attacker | AGNews | | | Yahoo! | | | SST2 | | | Amazon | | |
|---|---|---|---|---|---|---|---|---|---|---|---|---|---|
| | | Clean | Attack | Defense | Clean | Attack | Defense | Clean | Attack | Defense | Clean | Attack | Defense |
| Rapid | PWWS | | 2.081 | 1.356 | | 4.958 | 3.308 | | 0.529 | 0.588 | | 4.745 | 3.678 |
| | TF | 0.008 | 2.460 | 1.317 | 0.008 | 4.693 | 3.128 | 0.006 | 0.662 | 0.571 | 0.007 | 4.003 | 4.607 |
| | BAE | | 2.464 | 1.295 | | 5.194 | 4.053 | | 0.669 | 0.594 | | 4.350 | 4.403 |

Table 12: The execution efficiency of inferring clean examples, generating, and defending against adversarial examples.

most of the deep semantics remain within the adversarial example. Therefore, `Rapid`is able to rectify the example with fewer perturbations needed to search.

## D   DEPLOYMENT DEMO

We have created an anonymous demonstration of `Rapid`, which is available on Huggingface Space[9]. To illustrate the usage of our method, we provide two examples in Figure 5. In this demonstration, users can either input a new phrase along with a label or randomly select an example from a supplied dataset, to perform an attack, adversarial detection, and adversarial repair.

---

[9]`https://huggingface.co/spaces/anonymous8/RPD-Demo`

**Reactive Perturbation Defocusing for Textual Adversarial Defense**

Clarifications

○ This demo has no mechanism to ensure the adversarial example will be correctly repaired by RPD. The repair success rate is actually the performance reported in the paper (approximately up to 97%).

○ The adversarial example and repaired adversarial example may be unnatural to read, while it is because the attackers usually generate unnatural perturbations. RPD does not introduce additional unnatural perturbations.

○ To our best knowledge, Reactive Perturbation Defocusing is a novel approach in adversarial defense. RPD significantly (>10% defense accuracy improvement) outperforms the state-of-the-art methods.

○ The DeepWordBug is an unknown attacker to the adversarial detector and reactive defense module. DeepWordBug has different attacking patterns from other attackers and shows the generalizability and robustness of RPD.

**Natural Example Input**

Select a testing dataset and an adversarial attacker to generate an adversarial example.

○ SST2  ○ AGNews10K  ○ Amazon

Choose an Adversarial Attacker for generating an adversarial example to attack the model.

○ BAE  ● PWWS  ○ TextFooler  ○ DeepWordBug

Alternatively, input a natural example and its original label (from above datasets) to generate an adversarial example.

Input a natural example...

Original Label

Original label, must be an integer...

Generate an adversarial example to repair using RPD (GPU: < 1 minute, CPU: 1-10 minutes)

GPU status

Please click to check

Check if GPU available

**Generated Adversarial Example and Repaired Adversarial Example**

Original Example

anchored by a terrific performance by abbass , satin rouge shows that the idea of women 's self-actualization knows few continental divides .

Original Label

1

Adversarial Example

anchored by a terrific performance by abbass , satin rouge indicate that the estimate of women 's self-actualization screw few continental split .

Predicted Label of the Adversarial Example

0

Repaired Adversarial Example by RPD

anchored by a terrific performance by abbass , satin rouge indicate that the estimate of women 's self-actualization bang few continental split .

Predicted Label of the Repaired Adversarial Example

1

**Example Difference (Comparisons)**

The (+) and (-) in the boxes indicate the added and deleted characters in the adversarial example compared to the original input natural example.

📊 The Original Natural Example

anchored by a terrific performance by abbass , satin rouge shows that the idea of women 's self-actualization knows few continental divides .

📊 Character Editions of Adversarial Example Compared to the Natural Example

anchored by a terrific performance by abbass , satin rouge shows [-] indicate [+] that the est [+] i d [-] mat [+] e a [-] of women 's self-actualization k no [-] scre [+] w s [-] few continental d [-] spl [+] i t [+] vides [-] .

📊 Character Editions of Repaired Adversarial Example Compared to the Natural Example

anchored by a terrific performance by abbass , satin rouge shows [-] indicate [+] that the est [+] i d [-] mat [+] e a [-] of women 's self-actualization k [-] ba [+] n g [+] ows [-] few continental d [-] spl [+] i t [+] vides [-] .

**The Output of Reactive Perturbation Defocusing**

Adversarial Example Detection Result

| confidence | is_adversarial | perturbed_label |
|---|---|---|
| 1 | true | 0 |

The is_adversarial field indicates if an adversarial example is detected. The perturbed_label is the predicted label of the adversarial example. The confidence field represents the confidence of the predicted adversarial example detection.

Repaired Standard Classification Result

| confidence | is_correct | is_repaired | pred_label |
|---|---|---|---|
| 0.522 | Correct | true | 1 |

If is_repaired=true, it has been repaired by RPD. The pred_label field indicates the standard classification result. The confidence field represents the confidence of the predicted label. The is_correct field indicates whether the predicted label is correct.

Figure 5: The demo examples of adversarial detection and defense built on `Rapid` for defending against multi-attacks. The comparisons between natural and repaired examples are available based on the "*difflib*" library. The "$+$" and "$-$" in the colored boxes indicate letters addition and deletion compared to the natural examples. It is observed that `Rapid` only injects only one perturbation to repair the adversarial example, i.e., changing "screw" to "bang" in the adversarial example.

