# OpenReview forum: "The Best Defense is Attack: Repairing Semantics in Textual Adversarial Examples"
_ICLR.cc/2024/Conference — Submitted to ICLR 2024_

### Official Review · Reviewer_7Ywq · 2023-10-29

**Soundness:** 2 fair
**Presentation:** 2 fair
**Contribution:** 2 fair
**Rating:** 5
**Confidence:** 3

**Summary:**

This paper introduces a new defense framework, Rapid, designed to counter textual adversarial attacks. Rapid initially uses an adversarial example detector embedded within the victim model to identify potential adversarial examples. These flagged inputs are then further disturbed using adversarial attacks to neutralize the attacker's original perturbation, effectively restoring the adversarial example to a benign state. Lastly, a pseudo-similarity filtering strategy is implemented to select the restored examples, enhancing performance.

**Strengths:**

Rapid leverages adversarial attacks to counterattack adversaries and inject safe perturbations, aiming to distract the model from malicious perturbations.

**Weaknesses:**

1. Lacking evaluation on adaptive attack scenarios. The manuscript assumes a non-adaptive attacker without knowledge of the defense mechanism. However, modern attackers are becoming increasingly sophisticated and may attempt to circumvent defenses by targeting their individual components. Adding experiments where the adversarial detector and other defense modules are explicitly known to the attacker could demonstrate Rapid's effectiveness against such adaptive attacks.

2. While the joint training of the adversarial detector and victim model is an efficient approach, the manuscript could further analyze how this multi-task learning impacts the model's performance on natural examples. Specifically, it merits exploration of whether optimizing the additional loss terms introduced by Rapid has any detrimental effects on the core NLP capabilities that the model was originally designed for.

3. Lacking necessary ablation studies. The training losses listed in eq. 3 employs an adversarial training objective item. It is not clear if the defensive capability of Rapid is sourced from the existing adversarial training technique. Benchmarking against well-known adversarial training baselines would help evaluate the extent to which Rapid exceeds such prior work.

4. The paper asserts that Rapid outperforms existing methods in efficiency, as it only executes a Perturbation Defocusing step on potential AEs. However, the PD process, which involves adversarial attacks, appears to be time-consuming. To substantiate the claim of efficiency, the authors should provide experimental validation demonstrating cost efficiency on a dataset comprised of 50% clean samples and 50% successful AEs.

**Questions:**

Please refer to the Weaknesses sec.

---

> ### Author Response · Authors · 2023-11-18
> **Response to Reviewer 7Ywq [1/3]**
>
> **Weakness 1:**
> - **Lacking evaluation on adaptive attack scenarios.**
>   > **Reply:** We acknowledge and appreciate the reviewer’s insights regarding adaptive attack scenarios. However, the primary focus of this paper is twofold: first, to mitigate semantic bias in existing adversarial defenses, and second, to enhance the efficiency of detecting adversarial attacks. These are both pivotal aspects of textual adversarial defense.
>
>   > Our emphasis on efficiency stems from the observation that defense costs in practical systems are substantial. Thanks to Rapid's reactive defense mechanism, it proves to be highly efficient, especially when the proportion of adversarial examples is minimal compared to the volume of inputs, which can be in the millions. For detailed experimental results demonstrating the efficiency of our defense strategy, please refer to **Weakness 4**.
>
> - **The manuscript assumes a non-adaptive attacker without knowledge of the defense mechanism. However, modern attackers are becoming increasingly sophisticated and may attempt to circumvent defenses by targeting their individual components.**
>   > **Reply:** Our research primarily focuses on addressing the challenges of black-box defense, which means that the context of white-box defense is outside the context of our current study.
>
>   > Nonetheless, as part of our evaluation process for defending against black-box adaptive attacks, we have diligently assessed the performance of our methodology against a variety of unknown adversarial attacks. The results of these evaluations are presented in Table 4 in the submission. The results demonstrate that Rapid exhibits promising generalizability in identifying and repairing adversarial examples generated by PSO, IGA, DeepwordBug, and Clare, with performance levels that are comparable to those observed with unattacked original text. Additionally, Rapid's adversarial defense module is designed to be independent and compatible with existing textual adversarial detection methods.
>
>
>   > We acknowledge the complexities involved in detecting and defending against white-box adaptive textual attacks. Addressing and mitigating these adaptive attacks will be a focal point in our future research endeavors.
>
> **Weakness 2: While the joint training of the adversarial detector and victim model is an efficient approach, the manuscript could further analyze how this multi-task learning impacts the model's performance on natural examples. Specifically, it merits exploration of whether optimizing the additional loss terms introduced by Rapid has any detrimental effects on the core NLP capabilities that the model was originally designed for.**
>
> > **Reply:** Before developing Rapid, we carefully considered the potential impact on classification performance due to the multi-task training objectives. This consideration was explored in our proof-of-concept experiments.
>
> > To delve deeper into this impact, we trained victim models as single-task models, instead of multi-task training (i.e., no adversarial detection objective and adversarial training objective), and then collated detailed results for comparison with Rapid. In this experiment, we focused solely on evaluating performance using pure natural examples. The results of this comparison are outlined in the table below (Please find this table in the revised submission for better visualization). The symbols “↑” and “↓” accompanying the numbers indicate whether the performance is better or worse than that of the single-task model, respectively.
>
> > | Dataset | Model | Single-task Accuracy (%) | Multi-task Accuracy (%) |
> > |---------|-------|--------------------------|-------------------------|
> > | AGNews  | BERT  | **94.30**                    | 93.90 (−0.40↓)          |
> > | Yahoo!  | BERT  | 76.45                    | **76.61** (+0.16↑)          |
> > | SST2    | BERT  | **91.70**                    | 91.49 (−0.21↓)          |
> > | Amazon  | BERT  | **94.24**                    | **94.24** (—)               |
>
> > Based on these results, it is apparent that the inclusion of additional loss terms in the multi-task training objectives does impact the victim model's performance on natural examples. However, this influence is not substantial across all datasets and shows only slight variations. This finding suggests that the impact of the multi-task training objectives is relatively minor when compared to traditional adversarial training methods.
>
> > In addition, we conducted an ablation study of Rapid, excluding the adversarial training objective. This study was designed to isolate and assess the specific influence of the adversarial training objective. For further details on this aspect, please refer to our response to the **Weakness 4 by Reviewer EQRG**.

---

> ### Author Response · Authors · 2023-11-18
> **Response to Reviewer 7Ywq [2/3]**
>
> **Weakness 3:**
> - **Lacking necessary ablation studies. The training losses listed in eq. 3 employs an adversarial training objective item. It is not clear if the defensive capability of Rapid is sourced from the existing adversarial training technique.**
>   > **Reply:** In Rapid, the inclusion of adversarial training loss serves to enhance the effectiveness of adversarial detection and defense. The core module of our approach, however, is perturbation defocusing.
>
>   > To illustrate the impact of the adversarial training objective, we conducted ablation experiments. The results of these experiments are detailed in the following table (Please find this table in the revised submission for better visualization). We observed that the accuracy of adversarial defense decreases when the adversarial training loss is removed, suggesting a decline in adversarial defense performance. However, this impact on the performance of adversarial defense is relatively minor compared to its effect on adversarial detection. In conclusion, the adversarial training objective in Rapid proves to be highly effective and significantly enhances both adversarial detection and defense capabilities.
>
>   >|Defender|Attacker|AGNews(4-category)|||||Yahoo!(10-category)|||||SST2(2-category)|||||Amazon(2-category)|||||
>   >|--------------|----------|---------------------|-------|-------|-------|-------|----------------------|-------|-------|-------|-------|-------------------|-------|-------|-------|-------|---------------------|-------|-------|-------|-------|
>         >|||Nat.Acc.|Att.Acc.|Det.Acc.|Def.Acc.|Rep.Acc.|Nat.Acc.|Att.Acc.|Det.Acc.|Def.Acc.|Rep.Acc.|Nat.Acc.|Att.Acc.|Det.Acc.|Def.Acc.|Rep.Acc.|Nat.Acc.|Att.Acc.|Det.Acc.|Def.Acc.|Rep.Acc.|
>   >||PWWS||11.1|82.88|92.07|90.7||3.46|78.43|87.42|63.79||10.7|91.41|99.62|89.6||16.5|96.5|99.30|93.6|
>   >|Rapidw/oAT|TF|94.44|16.09|84.88|93.07|87.28|76.32|0.42|78.65|78.36|56.72|91.54|5.3|89.48|95.15|85.8|94.29|17.53|98.63|99.17|92.78|
>   >||BAE||67.93|83.17|91.49|91.15||45.1|71.89|75.47|64.56||25.7|57.01|95.64|87.1||45.54|92.67|99.48|93.31|
>   >||PWWS||32.09|90.11|95.88|92.36||5.70|87.33|92.47|69.40||23.44|94.03|98.62|89.85||15.56|97.33|99.99|94.42|
>   >|Rapid|TF|94.30|50.50|90.29|96.76|92.14|76.45|13.60|87.49|93.54|70.50|91.70|16.21|94.03|99.86|89.72|94.24|21.77|93.85|99.99|93.96|
>   >||BAE||74.80|57.55|96.25|93.64||27.50|82.46|96.30|73.06||35.21|78.99|99.28|89.77||44.00|80.55|99.99|93.89|
>
> - **Benchmarking against well-known adversarial training baselines would help evaluate the extent to which Rapid exceeds such prior work.**
>   > **Reply:** We have also conducted experiments to evaluate the performance of the adversarial training baseline (AT). The results, which compare AT to Rapid in Rep. Acc. (%), are presented in the table below.
>   > | Dataset | Attack | Rapid | AT |
>   > |---------|--------|-------|---------------|
>   > |   |  PWWS  | 92.36 |     60.10     |
>   > |  AGNews |   TF   | 92.14 |     61.87     |
>   > |         |  BAE   | 93.64 |     63.62     |
>   > |  |  PWWS  | 69.40 |     40.21     |
>   > | Yahoo!   |   TF   | 70.50 |     38.75     |
>   > |         |  BAE   | 73.06 |     42.97     |
>   > |    |  PWWS  | 89.85 |     32.46     |
>   > |   SST2 |   TF   | 89.72 |     31.23     |
>   > |         |  BAE   | 89.77 |     34.61     |
>   > |   |  PWWS  | 94.42 |     51.90     |
>   > | Amazon |   TF   | 93.96 |     49.49     |
>   > |         |  BAE   | 93.89 |     49.75     |
>
>   > For these experiments, we used BERT as the victim model and maintained the same experimental setup as for Rapid, including the number of adversaries used for training. It's important to note that we focus solely on the repaired accuracy metric, as AT does not facilitate detect-to-defense functionality. From these results, it becomes apparent that the traditional adversarial training baseline is less effective compared to Rapid, which utilizes perturbation defocusing. Specifically, the adversarial defense accuracy of AT is generally below 50%. This finding underscores the limitations of traditional adversarial training methods, particularly their high cost and reduced effectiveness against adapted adversarial attacks.

---

> ### Author Response · Authors · 2023-11-18
> **Response to Reviewer 7Ywq [3/3]**
>
> **Weakness 4: The paper asserts that Rapid outperforms existing methods in efficiency, as it only executes PD on potential AEs. However, the PD process, which involves adversarial attacks, appears to be time-consuming. To substantiate the claim of efficiency, the authors should provide experimental validation demonstrating cost efficiency on a dataset comprised of 50% clean samples and 50% successful AEs.**
> > **Reply:** The main efficiency depends on multiple adversarial perturbations search. We have implemented new experiments to investigate the efficiency of Rapid.
>
>   - **Time Costs for Multiple Examples:** We have collected three small sub-datasets that contain different numbers of adversarial examples and natural examples, say 200:0, 100:100, and 0:200. We apply adversarial detection and defense to this dataset and calculate the time costs. The results (measurement: second) are available in the following table (Please find this table in the revised submission for better visualization).
>
>     > | Attack | AGNews |  |  | Yahoo! |  |  | SST2 |  |  | Amazon |  |  |
>     > |--------|--------|-------|-------|-------|-------|-------|------|-------|-------|-------|-------|-------|
>     > |        | 200:0  | 100:100 | 0:200 | 200:0  | 100:100 | 0:200 | 200:0 | 100:100 | 0:200 | 200:0 | 100:100 | 0:200 |
>     > | PWWS   |        | 142.090 | 298.603 |       | 313.317 | 621.196 |      | 36.268 | 126.054 |      | 438.532 | 875.083 |
>     > | TF     | 1.188  | 146.654 | 293.542 | 1.157  | 314.926 | 642.206 | 1.092 | 51.303 | 137.795 | 1.138 | 329.075 | 665.052 |
>     > | BAE    |        | 141.434 | 260.231 |       | 352.186 | 876.606 |      | 52.626 | 138.325 |      | 349.256 | 655.264 |
>
>     > In this table, we can observe that Rapid's efficiency in handling adversarial examples varies across different attack methods and datasets, with a general trend of increasing time costs as the number of adversarial examples increases. The data suggests that while Rapid is effective, there is a non-negligible computational overhead associated with the adversarial defense, particularly when the dataset contains a high proportion of adversarial examples. However, Rapid will benefit from a large ratio of natural examples in practice, say millions of natural examples vs thousands of adversarial examples.
>
>   - **Time Costs for Single Examples:** We have also detailed the time costs per clean example, adversarial attack, and adversarial defense in PD. The experimental results can be found in the following table (Please find this table in the revised submission for better visualization). According to the experimental results, PD is slightly faster than the adversarial attack in most cases. Intuitively, the perturbed semantics in a malicious adversarial example are generally not robust, as most of the deep semantics remain within the adversarial example. Therefore, Rapid is able to rectify the example with fewer perturbations needed to search.
>
>     > | Defender | Attack | AGNews |  |  | Yahoo! |  |  | SST2 |  |  | Amazon |  |  |
>     > |----------|--------|--------|-------|--------|-------|-------|--------|-------|-------|--------|-------|-------|--------|
>     > |          |        | clean  | Attack | Defense | clean  | Attack | Defense | clean  | Attack | Defense | clean  | Attack | Defense |
>     > | Rapid    | PWWS   |          | 2.081  | 1.356   |           | 4.958  | 3.308   |           | 0.529  | 0.588   |           | 4.745  | 3.678   |
>     > |          | TF     | 0.008    | 2.460  | 1.317   | 0.008     | 4.693  | 3.128   | 0.006     | 0.662  | 0.571   | 0.007     | 4.003  | 4.607   |
>     > |          | BAE    |          | 2.464  | 1.295   |           | 5.194  | 4.053   |           | 0.669  | 0.594   |           | 4.350  | 4.403  |
>
>
>   > Please note that the time costs for adversarial attack and defense are dependent on specific software and hardware environments. These results are based on the experimental settings listed in the appendix.
>
> **Overall Response:**
> > We appreciate your help for your insights to help us improve our work, and the responses will be included in any future revision. But if you have further questions or require clarification on any of our responses, please do not hesitate to let us know. We will address and resolve any issues promptly.

---

> ### Author Response · Authors · 2023-11-22
>
> Dear Reviewer 7Ywq,
>
> We apologize for adding to your workload due to the review of our rebuttals.
>
> We have made significant efforts to address your concerns, given that the discussion period concludes today, we would greatly appreciate your review of our efforts and your valuable feedback. Your insights are invaluable in enhancing the quality of our work. We are deeply thankful for your detailed comments and your contributions to the open review community.
>
> Best regards

---

> > ### Comment · Reviewer_7Ywq · 2023-11-23
> > **Reponse to Rebuttal**
> >
> > I appreciate the dedication and hard work put in by the authors. However, I would like to highlight my ongoing concerns regarding the absence of evaluation on adaptive attack scenarios and Rapid's strong dependencies with existing adversarial training techniques. Due to these concerns, I have decided to maintain my current score.

---

> > > ### Author Response · Authors · 2023-11-23
> > >
> > > Thank you for your valuable feedback.
> > >
> > > We acknowledge the concern regarding the evaluation of adaptive adversarial defense in Rapid. Due to time constraints, a detailed evaluation of adaptive adversarial defense could not be provided in this discussion, but we intend to address this concern soon.
> > >
> > > **Regarding the confusion about the role of adversarial training techniques in Rapid, as highlighted in **_Weakness 3_**, we offer the following clarifications:**
> > >
> > > Final Metrics: The effectiveness of Rapid in textual adversarial detection and defense remains state-of-the-art, with no significant drop in performance even when the adversarial training objective is omitted. **In terms of the final metrics, Rapid is not strongly dependent on the adversarial training objective.**
> > >
> > > Attacked Accuracy: While the removal of adversarial training leads to a decrease in attacked accuracy on victim models, this does not adversely affect Rapid's performance in detecting and defending against textual adversarial attacks. Notably, **Rapid maintains its high efficacy even when initial adversarial robustness is limited**, as indicated by low attacked accuracy. This underscores Rapid's capability to counter adversarial attacks effectively.
> > >
> > > Adaption Potential of PD: The perturbation defocusing (PD) technique in Rapid demonstrates its utility across various scenarios, particularly when the victim model is significantly compromised by adversarial attacks. **This versatility reinforces the potential of integrating PD into existing and future adversarial defense techniques.**
> > >
> > > We hope these points adequately address your concerns regarding the contribution and significance of adversarial training in the context of Rapid. We appreciate your insightful review, and please have a nice day.
> > >
> > > Best regards.

---

### Official Review · Reviewer_EQRG · 2023-11-01

**Soundness:** 3 good
**Presentation:** 3 good
**Contribution:** 2 fair
**Rating:** 5
**Confidence:** 4

**Summary:**

Recent studies show language models are vulnerable to adversarial attacks. Current defence techniques struggle to repair semantics, limiting practical utility. A novel approach called Reactive Perturbation Defocusing (Rapid) uses an adversarial detector to identify pseudo-labels and leverage attackers to repair semantics. Experimental results show Rapid's effectiveness in various attack

**Strengths:**

+ The method is clearly presented and easy to follow
+ The demonstration is available

**Weaknesses:**

- The code is unavailable
- The novelty of work is limited
- Some designs need more justification. For instance, why do you train adversaries in adversarial training?

**Questions:**

1. Some well-known related textual adversarial attacks should be discussed, e.g., [1], [2], [3].
2. The novelty of the proposed method is hard to judge. It seems like it leverages another adversaries to change the attacked sentence to the sentence with the original semantics.
3. Some method designs require more justifications. For instance, in adversarial defense detection, it is unclear to me why do the authors train the adversaries.
4. In pseudo-similarity supervision, do you have a formal definition of the semantics of the sample? or is it just the label?

[1] Morris, J. X., Lifland, E., Yoo, J. Y., Grigsby, J., Jin, D., & Qi, Y. (2020). Textattack: A framework for adversarial attacks, data augmentation, and adversarial training in nlp. arXiv preprint arXiv:2005.05909.

[2] Li, J., Ji, S., Du, T., Li, B., & Wang, T. (2018). Textbugger: Generating adversarial text against real-world applications. arXiv preprint arXiv:1812.05271.

[3] Boucher, N., Shumailov, I., Anderson, R., & Papernot, N. (2022, May). Bad characters: Imperceptible nlp attacks. In 2022 IEEE Symposium on Security and Privacy (SP) (pp. 1987-2004). IEEE.

---

> ### Author Response · Authors · 2023-11-18
> **Response to Reviewer EQRG [1/3]**
>
> **Weakness 1: The code is unavailable**
> > **Reply:** The model and inference codes have been published alongside our online demo and submitted as supplementary material, and we've also made the code available on an anonymized GitHub repository (https://anonymous.4open.science/r/Rapid-ADCD) for ease of access. Furthermore, the data preprocessing and training codes will soon be released, in conjunction with the demo.
>
> > In a further commitment to open-source support, we have developed a user-friendly application interface. This interface is intended to act as a robust benchmarking tool for evaluating the performance of adversarial attackers within Rapid’s defense mechanism. It marks a significant stride towards reducing evaluation variance across different codebases. We plan to release this tool post-review process. The delay in release is attributable to the extensive workload involved in its development and the complexities of anonymizing the code.
>
> **Weakness 2: The novelty of work is limited**
> >**Reply:** We would like to justify our motivation from the following two aspects, which we believe advanced the state-of-the-art.
>
> > Enhancing Semantic Accuracy: Existing approaches often lack the supervision necessary to ensure accurate semantic repair of adversarial examples, as illustrated by the RS&V examples in Figures 1 and 4. To overcome this limitation, we introduce the pseudo-similarity supervision mechanism. This mechanism effectively filters out repaired examples that display biased semantics, thereby ensuring the repaired content maintains its original meaning. This feature distinguishes Rapid from other methods in terms of effectiveness and semantic fidelity.
>
> > Improving Defense Performance: Many existing adversarial defense methods show limited performance. In contrast, Rapid employs adversarial attackers for the active repair of adversarial examples based on their predicted labels. This novel application of adversarial attackers in defense strategies is both innovative and effective within the scope of textual adversarial defense. It provides a unique perspective on formulating and expediting adversarial defense methods.
>
> > In summary, Rapid introduces a perspective to adversarial defense methodologies by utilizing adversarial attackers to accurately repair adversarial examples.

---

> ### Author Response · Authors · 2023-11-18
> **Response to Reviewer EQRG [2/3]**
>
> **Weakness 3: Some designs need more justification. For instance, why do you train adversaries in adversarial training?**
> > **Reply:** The rationale behind the adversarial training objective in our study is founded on two key hypotheses.
> > Improving Model Robustness: We posit that an adversarial training objective can bolster the model's robustness, thereby mitigating performance degradation when the model faces an attack. This approach is designed to strengthen the model against potential adversarial threats.
>
> > Enhancing Adversarial Detection: We recognize an implicit link between the tasks of adversarial training and adversarial example detection. Our theory suggests that by incorporating an adversarial training objective, we can indirectly heighten the model's sensitivity to adversarial examples, leading to more accurate detection of such instances.
>
> > To validate these hypotheses, we conducted ablation experiments on the adversarial training objective. The experimental setup was aligned with that described in Table 2 in the submission, and the results are outlined in the following table (please find the table in the updated submission for better visualization):
>
>
> > | Defender     | Attacker | AGNews (4-category) |       |       |       |       | Yahoo! (10-category) |       |       |       |       | SST2 (2-category) |        |       |       |       | Amazon (2-category) |       |       |       |       |
> > |--------------|----------|---------------------|-------|-------|-------|-------|----------------------|-------|-------|-------|-------|-------------------|-------|-------|-------|-------|---------------------|-------|-------|-------|-------|
>  |              |          | Nat. Acc.           | Att. Acc. | Det. Acc. | Def. Acc. | Rep. Acc. | Nat. Acc.            | Att. Acc. | Det. Acc. | Def. Acc. | Rep. Acc. | Nat. Acc.           | Att. Acc. | Det. Acc. | Def. Acc. | Rep. Acc. | Nat. Acc.           | Att. Acc. | Det. Acc. | Def. Acc. | Rep. Acc.  |
> > |  | PWWS     |                     | 11.1  | 82.88 | 92.07 | 90.7  |                     | 3.46  | 78.43 | 87.42 | 63.79 |                     | 10.7  | 91.41 | 99.62 | 89.6  |                     | 16.5  | 96.5  | 99.30 | 93.6  |
> > | Rapid w/o AT  | TF       | 94.44               | 16.09 | 84.88 | 93.07 | 87.28 | 76.32               | 0.42  | 78.65 | 78.36 | 56.72 | 91.54             | 5.3   | 89.48 | 95.15 | 85.8  | 94.29             | 17.53 | 98.63 | 99.17 | 92.78 |
> > |              | BAE      |                     | 67.93 | 83.17 | 91.49 | 91.15 |                     | 45.1  | 71.89 | 75.47 | 64.56 |                     | 25.7  | 57.01 | 95.64 | 87.1  |                     | 45.54 | 92.67 | 99.48 | 93.31 |
>
> > | Defender     | Attacker | AGNews (4-category) |       |       |       |       | Yahoo! (10-category) |       |       |       |       | SST2 (2-category) |        |       |       |       | Amazon (2-category) |       |       |       |       |
> > |--------------|----------|---------------------|-------|-------|-------|-------|----------------------|-------|-------|-------|-------|-------------------|-------|-------|-------|-------|---------------------|-------|-------|-------|-------|
> > |              |          | Nat. Acc.           | Att. Acc. | Det. Acc. | Def. Acc. | Rep. Acc. | Nat. Acc.            | Att. Acc. | Det. Acc. | Def. Acc. | Rep. Acc. | Nat. Acc.           | Att. Acc. | Det. Acc. | Def. Acc. | Rep. Acc. | Nat. Acc.           | Att. Acc. | Det. Acc. | Def. Acc. | Rep. Acc. |
> > |                | PWWS     |                     | 32.09 | 90.11 | 95.88 | 92.36 |                     | 5.70  | 87.33 | 92.47 | 69.40 |                     | 23.44 | 94.03 | 98.62 | 89.85 |                     | 15.56 | 97.33 | 99.99 | 94.42 |
> > |   Rapid       | TF       | 94.30               | 50.50 | 90.29 | 96.76 | 92.14 | 76.45               | 13.60 | 87.49 | 93.54 | 70.50 | 91.70             | 16.21 | 94.03 | 99.86 | 89.72 | 94.24             | 21.77 | 93.85 | 99.99 | 93.96 |
> > |              | BAE      |                     | 74.80 | 57.55 | 96.25 | 93.64 |                     | 27.50 | 82.46 | 96.30 | 73.06 |                     | 35.21 | 78.99 | 99.28 | 89.77 |                     | 44.00 | 80.55 | 99.99 | 93.89 |
>
> > These experimental findings reveal that omitting the adversarial training objective in Rapid consistently leads to a reduction in model robustness across all datasets. This reduction can be as substantial as approximately 30%, adversely affecting the performance of the adversarial defense. Additionally, adversarial detection capabilities also diminish, with the most significant drop being around 20%. These results highlight the critical role of the adversarial training objective in Rapid, confirming its efficacy in enhancing both model robustness and adversarial example detection capabilities.

---

> ### Author Response · Authors · 2023-11-18
> **Response to Reviewer EQRG [3/3]**
>
> **Question 2:**
>   - **The novelty of the proposed method is hard to judge.**
>   > **Reply:** We have justified the main novelty in Weakness 2.
>   - **It seems like it leverages another adversaries to change the attacked sentence to the sentence with the original semantics.**
>   > **Reply:** Yes, this is an unusual usage of adversarial attackers which achieves state-of-the-art adversarial defense performance on all datasets.
>
> **Question 3: Some method designs require more justifications. For instance, in adversarial defense detection, it is unclear to me why do the authors train the adversaries.**
>   > **Reply:** We have conducted new experiments to demonstrate the impact of the adversarial training objective through ablation studies.
>
>   > The main motivation for training the adversarial detector based on sampled adversarial examples is that we found the language models are very sensible of adversarial examples, and this finding can be used for training an adversarial detector on the victim model based on multi-task learning. More specifically,  to address the inefficiencies associated with adversarial detection, Rapid employs an 'in-victim-model' adversarial detector. This detector is a binary classifier built upon the Pre-trained Language Model (PLM) architecture. It is jointly trained with the victim model as part of a multitask modeling approach, enabling it to detect adversaries without incurring additional costs. Practically, this adversarial detector is capable of recognizing adversaries generated by a variety of attackers. This capability stems from its training on a diverse set of adversarial examples originating from multiple adversarial attackers.
>
> **Question 4: In pseudo-similarity supervision, do you have a formal definition of the semantics of the sample? or is it just the label?**
>   > **Reply:** We have not redefined the term “semantics.” Consistent with conventional understanding, we consider the semantics of an adversarial example to be the feature representation encoded by the victim model. This representation is a tensor, instead of a label.
>
>   > Pseudo-similarity supervision is employed to filter out examples with altered semantics. In Rapid, we initially use perturbation to determine the predicted labels and semantics of several repaired examples using the joint model. Subsequently, we calculate similarity scores for each example relative to the rest of the examples. We then compute the average similarity score for each example and exclude those with scores lower than the average. Ultimately,  Rapid outputs the repaired example (with its predicted label) which has the largest average similarity score.
>
>   > We will revise our manuscript to further clarify the process and purpose of pseudo-similarity supervision.
>
> **Overall Response:**
>  > We appreciate your help for your insights to help us refine this work, we will update the submission soon and the responses will be incorporated into any future revisions. But if you have further questions or require clarification on any of our responses, please do not hesitate to reach out. We will address and resolve any issues that may arise.

---

> ### Author Response · Authors · 2023-11-22
>
> Dear Reviewer EQRG,
>
> We apologize for adding to your workload due to the review of our rebuttals.
>
> We have made significant efforts to address your concerns, given that the discussion period concludes today, we would greatly appreciate your review of our efforts and your valuable feedback. Your insights are invaluable in enhancing the quality of our work. We are deeply thankful for your detailed comments and your contributions to the open review community.
>
> Best regards

---

### Official Review · Reviewer_WkfB · 2023-11-01

**Soundness:** 3 good
**Presentation:** 2 fair
**Contribution:** 2 fair
**Rating:** 6
**Confidence:** 2

**Summary:**

The paper proposes Rapid, a method to correct adversarial perturbations in textual adversarial examples.

Rapid is a reactive adversarial defense, doesn't defend all inputs. It pre-detects adversaries using an adversarial detector which is jointly trained with a standard classifier. Once adversarial input is identified, safe perturbations are added (perturbation defocusing). The perturbation introduced by the adversarial attacker is considered ‘safe’ since it does not alter the semantics of the adversarial input. *The safe perturbations are also added by an adversarial attack method.*

The paper shows that recent adversarial defense RS&V [A] cannot model the semantic differences in adversarial and repaired examples. Moreover, prior work is unable to efficiently pre-detect adversaries before the defense process. These approaches indiscriminately treat all input texts.

[A] Xiaosen Wang, Yifeng Xiong, and Kun He. Detecting textual adversarial examples through randomized substitution and vote. In UAI, volume 180 of Proceedings of Machine Learning Research, pp. 2056–2065. PMLR, 2022b.

Experiments are performed on BERT and DeBERTa models on SST2, Amazon, Yahoo! and AGNews text classification datasets.
Multiple baseline defense methods like DISP, FGWS and RS&V are compared against Rapid.

**Strengths:**

The paper shows experimental results which improve upon prior defense works like DISP, FGWS and RS&V.
Three research questions study the behavior of the defense method in detail.
The working demo illustrates the inputs and outputs of the system well.

**Weaknesses:**

The use of adversarial attackers to repair adversarial perturbations by Rapid is a bit counterintuitive to me. I do not fully understand how Rapid can correct attacks generated by PWWS when it also internally uses PWWS (Table 2). Either the method section doesn't clearly convey the idea, or it's due to my unfamiliarity with prior work.

**Questions:**

1. (Typo) Section 7 “alailable”

2. (Clarification) Section 1 “The examples repaired by Rapid are well-maintained”. A clarification regarding what well-maintained means will be helpful.

---

> ### Author Response · Authors · 2023-11-18
> **Response to Reviewer WkfB [1/1]**
>
> **Weakness 1:**
>
> - **The use of adversarial attackers to repair adversarial perturbations by Rapid is a bit counterintuitive to me.**
>
>     > **Reply:** The motivation for our proposed perturbation defocusing (PD) approach stems from two key observations in the current literature.
>
>     > Existing approaches often lack the supervision needed to guarantee that the semantics of adversarial examples are accurately repaired, as evidenced by the RS&V examples in Figures 1 and 4. To remedy this, we introduce a pseudo-similarity supervision mechanism that effectively filters out repaired examples with biased semantics. This ensures the restored content maintains its original meaning, distinguishing Rapid in terms of both its effectiveness and its ability to preserve semantic fidelity.
>
>     > Current adversarial defense methods typically exhibit limited performance. Rapid innovatively employs adversarial attackers to actively repair adversarial examples based on their predicted labels. This reversal of the conventional role of adversarial attackers in defense strategies is novel and proves to be effective in the realm of textual adversarial defense, providing a new perspective for formulating and expediting adversarial defense strategies.
>
> - **I do not fully understand how Rapid can correct attacks generated by PWWS when it also internally uses PWWS (Table 2). Either the method section doesn't clearly convey the idea, or it's due to my unfamiliarity with prior work.**
>
>     > **Reply (Part 1):** Thank you for your insightful question regarding Rapid's use of the PD strategy. We will first explain why Rapid can repair adversarial examples and then justify why PWWS is effective within PD for defending against PWWS attacks.
>
>     > We discovered that the malicious perturbations in adversarial examples are not robust, and minimal changes can deactivate these perturbations and repair the examples. Thus, Rapid's core aim is to find a means of introducing such benign changes (i.e., "safe perturbations" when they rectify the examples) into adversarial inputs. We selected PWWS for this role in the PD process due to its balanced performance and efficiency, as detailed in Footnote 3 of our paper.
>
>
>     > As for why PWWS is effective, PD can incorporate any adversarial attacker or even an ensemble of attackers, as the process doesn't require prior knowledge of the specific malicious perturbations. Regardless of which adversaries are deployed against Rapid, PWWS consistently seeks safe perturbations for the current adversarial examples. The abstract nature of PD is critical, allowing for adaptability and effectiveness against a broad spectrum of adversarial attacks, rendering it a versatile defense mechanism in our study.
>
>
>     > Furthermore, defense is not simply the reverse operation of attack in terms of textual operations. Our observations show that safe perturbations identified by PWWS are not directly linked to malicious perturbations. This was illustrated through two distinct cases: one where both the attacker and defender use PWWS, and another with different attackers (TextFooler for the attacker and PWWS for the defender). In both scenarios, PD effectively mitigated incorrect predictions despite different adversarial approaches. In the first case (https://anonymous.4open.science/r/Rapid-ADCD/pwws-pwws.png), although both parties used the same PWWS strategy, PD-generated safe perturbations were unrelated to the malicious ones. In the second case (https://anonymous.4open.science/r/Rapid-ADCD/textfooler-pwws.png), different strategies led to the same incorrect prediction by the victim model (9 → 5), yet PD successfully corrected both adversarial examples (5 → 9), showcasing the efficacy of PD across various adversarial tactics.

---

> ### Author Response · Authors · 2023-11-18
> **Response to Reviewer WkfB [2/2]**
>
> > **Reply (Part 2):**  Additionally, we have implemented further experiments to demonstrate the adversarial defense performance of PD using different attackers, namely TextFooler and BAE. The asterisk "*" denotes experiments based on a randomly selected subset containing 1,000 adversarial examples. The results are shown in the three tables below, where only the defense accuracy and repaired accuracy metrics apply to PD.
>
> > According to the experimental results, it is observed that PWWS has a similar performance to TextFooler in PD, while BAE is slightly inferior to both PWWS and TextFooler. However, the variance is not significant among different attackers in PD, which means the performance of Rapid is not sensitive to the choice of the adversarial attacker in PD, in contrast to the performance of the adversarial attack of the adversarial attacker. Please find the table 7 in the revision for better visualization.
> > | Defender     | Attacker | AGNews (4-category) |       | Yahoo! (10-category) |       | SST2 (2-category) |       | Amazon (2-category) |       |
> > |--------------|----------|---------------------|-------|----------------------|-------|-------------------|-------|---------------------|-------|
> > |   |   | Def. Acc.      | Rep. Acc. | Def. Acc.            | Rep. Acc. | Def. Acc. | Rep. Acc. | Def. Acc.     | Rep. Acc. |
> > |   | PWWS     | 95.88               | 92.36     | 92.47      | 69.40     | 98.62      | 89.85     | 99.99       | 94.42     |
> > | Rapid (PWWS) | TF     | 96.76      | 93.14     | 93.54      | 70.50     | 99.86     | 89.72     | 99.99     | 93.96     |
> > |  | BAE      | 96.25    | 92.64     | 96.30       | 73.06     | 99.28         | 89.77     | 99.99   | 93.89     |
>
>
>
> > | Defender     | Attacker | AGNews (4-category) |       | Yahoo! (10-category) |       | SST2 (2-category) |       | Amazon (2-category) |       |
> > |--------------|----------|---------------------|-------|----------------------|-------|-------------------|-------|---------------------|-------|
> > |              |          | Def. Acc.           | Rep. Acc. | Def. Acc.            | Rep. Acc. | Def. Acc.            | Rep. Acc. | Def. Acc.            | Rep. Acc. |
> > |              | PWWS     | 94.07     | 92.27     | 83.25   | 65.33     | 98.90  | 90.67     | 99.99                | 94.33      |
> > | Rapid (TF*)  | TF       | 96.46   | 92.67     | 92.96   | 71.00     | 99.98      | 90.73     | 99.99                | 94.33     |
> > |              | BAE      | 92.68   | 91.00     | 86.49    | 72.67     | 99.98  | 91.33     | 99.99                | 94.33     |
>
>
>
> > | Defender     | Attacker | AGNews (4-category) |       | Yahoo! (10-category) |       | SST2 (2-category) |       | Amazon (2-category) |       |
> > |--------------|----------|---------------------|-------|----------------------|-------|-------------------|-------|---------------------|-------|
> > |              |          | Def. Acc.           | Rep. Acc. | Def. Acc.            | Rep. Acc. | Def. Acc.            | Rep. Acc. | Def. Acc.            | Rep. Acc. |
> > |              | PWWS     | 93.22               | 92.08     | 81.15                | 64.00     | 93.92                | 87.67     | 99.54                | 94.00      |
> > | Rapid (BAE*) | TF       | 95.96               | 92.33     | 87.79                | 67.33     | 96.55                | 89.00     | 99.61                | 93.64     |
> > |              | BAE      | 95.12               | 91.33     | 83.78                | 72.00     | 97.55                | 90.00     | 99.15                | 93.80     |
>
>   > In our future work, we aim to further explore the relationship between the performance of source adversarial attackers and the attackers used for PD. This investigation is crucial for a deeper understanding of the interplay between adversarial attacks and defenses.
>
> **Question 1: (Typo) Section 7 “alailable”**
>   > **Reply:** We are currently proofreading this manuscript to eliminate any typos and errors.
>
> **Question 2: (Clarification) Section 1 “The examples repaired by Rapid are well-maintained”. A clarification regarding what well-maintained means will be helpful.**
>   > **Reply:** The statement in Section 1 could be further clarified as follows: “The semantics of the examples repaired by the victim model closely resemble those of the original, natural examples, as evidenced by Figure 1.” This similarity is underscored by the high cosine similarity scores between the adversarial examples and the repaired examples, which approach a value close to 1.
>
> **Overall Response:**
>   >  We greatly appreciate your insightful feedback, which will be instrumental in refining our work. We will update the submission soon and the responses will be incorporated into any future revisions. If you have further questions or require clarification on any of our responses, please do not hesitate to reach out. We are committed to addressing and resolving any issues that may arise.

---

> ### Author Response · Authors · 2023-11-22
>
> Dear Reviewer WkfB,
>
> We apologize for adding to your workload due to the review of our rebuttals.
>
> We have made significant efforts to address your concerns, given that the discussion period concludes today, we would greatly appreciate your review of our efforts and your valuable feedback. Your insights are invaluable in enhancing the quality of our work. We are deeply thankful for your detailed comments and your contributions to the open review community.
>
> Best regards

---

> > ### Comment · Reviewer_WkfB · 2023-11-22
> > **Response to Rebuttal**
> >
> > I thank the authors for their detailed response.
> >
> > I have no additional questions.

---

> > > ### Author Response · Authors · 2023-11-22
> > >
> > > Thank you very much for your feedback and patience in improving this work!

---

### Author Response · Authors · 2023-11-21
**Discussion Participation Request**

Dear All Reviewers and Area Chairs,

We are deeply appreciative of your patience and efforts towards improving this work.

We are looking forward to addressing any remaining questions from the reviewers and ACs. Please let us know if you have any additional inquiries. Your further feedback or insights on our responses are highly valued and we are grateful for them.

Kind regards,

Authors of Submission 1263

---

### Author Response · Authors · 2023-11-22
**Revision Summary**

Dear All Reviewers and Area Chairs,

We are grateful for your comments to help us improve our work and thanks for your contribution to the open review community.

We have uploaded a revision of the submission which includes most of the responses. Here are the main updates:

1. We have added all new ablation experiments and results from all reviewers in Appendix C, such as the training objectives ablation experiments and adversarial training baseline experiments, efficiency evaluation experiments, etc.

2. We have clarified the confusing statements, such as the pseudo-similarity supervision-based repaired examples filtering.

3. We have included the missing references provided by Reviewer EQRG. Moreover, we have provided a brief investigation of many textual adversarial attacks in Appendix A to help readers understand the development of adversarial attacks.

4. We have released the code in a more accessible location, which can help reproduce the performance of adversarial defense in a few lines of code.

Thank you all for your insightful comments. Your further feedback or questions on our responses are highly valued.

Kind regards,

Authors

---

### Meta-Review · Area_Chair_LAJd · 2023-12-08

**Metareview:**

This paper proposes a defense against textual adversarial examples that combines an attack detector with an extra step to repair the semantics of the input.

Defending against textual adversarial examples is a hard and important problem, so such approaches are needed and timely.
Unfortunately, we've also learned from countless defenses against adversarial examples in the vision space that defenses that don't account for and evaluate on adaptive attacks are of limited use.

**Justification For Why Not Higher Score:**

The main drawback is the lack of evaluation on adaptive attacks. The authors acknowledge this in the rebuttal, and give additional results on held-out black-box attacks. But this is not an adaptive evaluation. These attacks are still oblivious to the defense.
We have countless evidence in this field already of defenses that are subsequently broken by adaptive attacks. At this point I expect defense papers to do more on this front.

**Justification For Why Not Lower Score:**

N.A

---

### Decision · Program_Chairs · 2024-01-16

Reject